# *Sox9* prevents retinal degeneration and is required for limbal stem cell differentiation in the adult mouse eye

Alicia Hurtado[1†§], Victor López-Soriano[1†#], Miguel Lao[1†¶],
M Angeles Celis-Barroso[1], Pilar Lazúen[1], Alejandro Chacón-de-Castro[1],
Yolanda Ramírez-Casas[1**], Miguel Alaminos[2,3], John Martin Collinson[4],
Miguel Burgos[1], Rafael Jiménez[1*‡], F David Carmona[1,2*‡],
Francisco Javier Barrionuevo[1*‡]

[1]Departamento de Genética e Instituto de Biotecnología, Centro de Investigación Biomédica (CIBM), Universidad de Granada, Granada, Spain; [2]Instituto de Investigación Biosanitaria ibs.GRANADA, Granada, Spain; [3]Departamento de Histología, Universidad de Granada, Granada, Spain; [4]School of Medicine, Medical Sciences and Nutrition, University of Aberdeen, Institute of Medical Sciences, Aberdeen, United Kingdom

**\*For correspondence:**
rjimenez@ugr.es (RJ);
dcarmona@ugr.es (FDC);
fjbarrio@go.ugr.es (FJB)

†These authors contributed equally to this work
‡These authors also contributed equally to this work

**Present address:** §Centro Andaluz de Biología del Desarrollo (CABD), CSIC/UPO/JA, Seville, Spain; #Internal Medicine V, Hematology, Oncology and Rheumatology, Heidelberg University Hospital, Heidelberg, Germany; ¶IBiS Instituto de Biomedicina de Sevilla, Seville, Spain; **Centro de Investigación Biomédica, Facultad de Medicina, Departamento de Fisiología, Instituto de Biotecnología, Granada, Spain

**Competing interest:** The authors declare that no competing interests exist.

## eLife Assessment

This **useful** study informs the transcriptional mechanisms that promote stem cell differentiation and prevent degeneration in the adult eye. Through inducible mouse mutagenesis, the authors uncover a dual role for a transcription factor (Sox9) in stem cell differentiation and prevention of retinal degeneration. The data at hand **convincingly** support to the main conclusions. The study will be of general interest to the fields of neuronal development and neurodegeneration.

**Abstract** *Sox9* is a transcription factor with multiple roles during development and in adult organ homeostasis. In the adult eye, *Sox9* expression persists in several cell types, including the retinal pigmented epithelium cells and the Müller glial (MG) cells, as well as in the limbal and corneal basal epithelia. To uncover the role of *Sox9* in these cell types, we induced the deletion of the gene in adult mice. We found that, after *Sox9* ablation, mutant mice undergo a severe process of retinal degeneration characterized by the loss of MG cells and complete depletion of the photoreceptors layer. Moreover, by combining single-cell RNA sequencing and *Sox9* lineage tracing, we found that *Sox9* is expressed in a basal limbal stem cell population with the ability to form two types of long-lived cell clones involved in stem cell maintenance and homeostasis. Mosaic analysis of *Sox9* positive and negative cells confirmed that the gene is essential for limbal stem cell differentiation. Our results show that *Sox9* is required for the maintenance of retinal integrity and for limbal stem cell differentiation in the adult mouse eye.

## Introduction

The retina and cornea represent indispensable components of the vertebrate visual system that ensure proper vision (*Casey et al., 2023*). The retina, a multilayered sensory tissue at the back of the eye, converts light into neural signals, initiating visual perception. It consists of various specialized cells organized into three nuclear layers and two synaptic layers. Photoreceptors, including rods and

cones, are located in the outer nuclear layer (ONL) and are responsible for converting light stimuli into electrical signals. These signals are then transmitted to bipolar cells in the inner nuclear layer (INL), which, in turn, establish synapses with retinal ganglion cells, whose axons form the optic nerve that sends the electrical impulse to the brain. Additionally, horizontal and amacrine cells, also positioned in the INL, provide lateral connections, offering feedback and feedforward signals. Finally, Müller glial (MG) cells, with nuclei located in the INL and bodies extending across the entire retina, provide structural, metabolic, and functional support to the other retinal cells, thus maintaining retinal function and integrity (*Baden et al., 2020*).

The cornea, positioned at the forefront of the eye, is a transparent and refractive structure responsible for transmitting and focusing light onto the retina, which is devoid of blood vessels. The outermost layer of the cornea is a stratified squamous epithelium that undergoes continuous regeneration. Stem cells responsible for this replenishment reside within a specialized niche located at the periphery, adjacent to the conjunctiva, known as the limbus (*Lavker et al., 2004*; *West et al., 2015*). Within this limbal niche, long-lived limbal epithelial stem cells (LESCs) give rise to a more differentiated progeny responsible for the regeneration of the entire corneal epithelium throughout two distinct pathways of cell migration: one within the basal layer, moving from the limbus to the center of the cornea, and another from the basal layer to the terminally differentiated, desquamated cells at the epithelial surface (*Yazdanpanah et al., 2017*).

*SOX9* is a transcription factor involved in the regulation of multiple developmental processes (*Jo et al., 2014*). In humans, mutations affecting *SOX9* cause campomelic dysplasia (CD), a syndrome characterized by skeletal malformations and XY sex reversal (*Wagner et al., 1994*; *Foster et al., 1994*). In the adult mice, SOX9 has been shown to play a role in the homeostasis of neural, liver, pancreas, intestine, and nail stem cells (*Scott et al., 2010*; *Furuyama et al., 2011*; *Lao et al., 2022*), as well as in the maintenance of Sertoli cells (*Barrionuevo et al., 2016*), among others.

Several works have addressed the role of *Sox9* during the developing eye. In the context of the mouse retina, *Sox9* is expressed in multipotent mouse retinal progenitor cells, undergoes downregulation during neuronal differentiation, and then the gene is exclusively expressed in MG cells and retinal pigment epithelia (RPE) until adulthood (*Poché et al., 2008*; *Muto et al., 2009*). Deletion of *Sox9* in developing retinal cells using a Chx10-Cre mouse line revealed that *Sox9*-mutant retinae exhibited normal lamination as well as nuclear layer thickness comparable to control samples, indicating that loss of *Sox9* alone likely does not affect global retinal fate determination (*Poché et al., 2008*). shRNA-induced downregulation of *Sox9* and *Sox8* in retinal explants resulted in a reduced population of MG cells and a moderate increase in the proportion of rod photoreceptors, indicating that both SoxE genes play roles in the specification of MG cells (*Muto et al., 2009*). Similar results were obtained when *Sox9* was conditionally deleted in the developing retina using a *Pax6*-Cre mouse line (*Goto et al., 2018*). In the mature RPE, *Sox9* collaborates with other transcription factors, such as OTX2 and LHX, to regulate the visual cycle, a series of biochemical reactions that regenerate 11-cis-retinal, the chromophore essential for phototransduction in photoreceptor cells (*Masuda et al., 2014*). Three studies have investigated *Sox9* inactivation in RPE cells. While one study reported no obvious morphological abnormalities in the retina and RPE upon conditional deletion of the gene (*Masuda et al., 2014*), two subsequent studies demonstrated that *Sox9* was required for proper choroid differentiation (*Goto et al., 2018*; *Cohen-Tayar et al., 2018*).

In the cornea, transcriptome analysis from slow cycling cells identified *Sox9* as a putative stem cell marker or determinant, and immunohistochemical analyses confirmed that the gene is expressed in LESCs and in corneal epithelial cells (*Sartaj et al., 2017*; *Parfitt et al., 2015*; *Menzel-Severing et al., 2018*). RNAi silencing of the gene in cultured limbal stem cells resulted in reduced expression of progenitor cell markers and increased expression of differentiation markers. Further analyses showed that *Sox9* and Wnt/β-catenin signaling antagonizes to maintain a balance between quiescence, proliferation, and differentiation of limbal stem cells (*Menzel-Severing et al., 2018*).

Although several studies have addressed the role of *Sox9* in retinal and corneal development (*Poché et al., 2008*; *Muto et al., 2009*; *Masuda et al., 2014*; *Sartaj et al., 2017*; *Parfitt et al., 2015*; *Menzel-Severing et al., 2018*), its specific in vivo contributions to the maintenance and regeneration of these tissues in adult mammals remain unknown.

In this work, we used a tamoxifen-inducible Cre/LoxP system to inactivate the *Sox9* gene in adult mice and a lineage-tracing mouse strain to study the fate of *Sox9*-expressing cells. With these tools,

in conjunction with scRNA-seq analysis, we conducted a comprehensive study of morphological changes, cell-specific marker expression, and lineage tracing to determine the role of *Sox9* in maintaining retinal integrity and corneal stem cell homeostasis.

## Results

### Deletion of *Sox9* leads to retinal degeneration in adult mice

To evaluate the role of *Sox9* in adult retinal homeostasis, we conditionally deleted the gene in adult mouse cells using a ubiquitous tamoxifen-inducible CAGG-CreER recombinase (*Hayashi and McMahon, 2002*) and a conditional *Sox9flox/flox* allele (; *Figure 1A*). Histological examination of retinas from control (Cre-negative *Sox9flox/flox*; *n* = 7) and *CAGG-CreER;Sox9flox/flox* (hereafter *Sox9Δ/Δ*; *n* = 24) adult mice (2 months old) at different days after TX administration (DATX) revealed that most of the analyzed mutant samples (*n* = 18/24; *Supplementary file 1A*) showed either no relevant morphological defects in the retinal sections or a mild phenotype in which the main retinal nuclear layers (INL and ONL) occupied a smaller area and their edges did not appear as clearly differentiated as in the control retinas. However, we also observed an extreme phenotype in a reduced number of mutant retinas (*n* = 6/24), characterized by the complete loss of the ONL (*Figure 1B*). We categorized *Sox9Δ/Δ* retinas into 'mild' and 'extreme' phenotypes in order to facilitate interpretation of our data. Classification was based on a qualitative assessment of ONL integrity in histological sections. Specifically, samples were classified as 'extreme' when the ONL was completely depleted, and as 'mild' when the ONL persisted, albeit variably reduced in thickness. This phenotypic classification reflects observable structural differences rather than a fixed quantitative threshold. Some variability exists within the 'mild' group, likely due to differences in recombination efficiency and the mosaic nature of tamoxifen-induced Cre-mediated *Sox9* deletion. The severity of the phenotype showed no correlation with the time elapsed after tamoxifen administration, as we found mutant retinas with no apparent phenotype at 100 DATX and others with an extreme phenotype at 20 DATX (*Supplementary file 1A*). Retinal degeneration was never observed in mice that had not been tamoxifen-treated, nor any other control groups, making the presence of the retinal degeneration allele of photoreceptor cGMP phosphodiesterase 6b (*Pde6brd1*) unlikely in our mice (*Bowes et al., 1990*). However, we acknowledge that definitive exclusion of this possibility would require PCR-based genotyping.

During retinal development, *Sox9* is expressed in neural RPCs, but once the neural retina is differentiated, *Sox9* expression is restricted to MG and RPE cells (*Poché et al., 2008*; *Masuda et al., 2014*). This is, indeed, the SOX9 expression pattern we observed in retinal sections obtained from adult control mice (*Figure 1C*). To explain the phenotypic diversity shown by the TX-treated mice, we assessed the efficiency of *Sox9* deletion in mutant mice. For this, we performed SOX9/SOX8 double immunofluorescence, as both transcription factors are normally co-expressed in MG cells (*Figure 1C*; *Muto et al., 2009*), and we consistently observed the presence of many SOX8+ cells that lacked SOX9 expression in the retinas of mutant TX-treated mice (*Figure 1C*). We calculated the percentage of SOX8+ cells that also co-expressed SOX9 in control and in *Sox9Δ/Δ* retinas with mild and extreme phenotypes (*Figure 1D*; *Supplementary file 1A and B*). In control MG cells, all SOX8+ cells also expressed SOX9. In contrast, in mutants with a mild phenotype, the average percentage was 55%, although with a large standard deviation (55 ± 23%), while in extreme mutants, the percentage was significantly lower (16 ± 12%). Altogether, these results indicate that the severity of the phenotype is directly related to the efficiency of *Sox9* inactivation, and that only individuals with *Sox9* inactivation occurring in at least 80% of the cells expressing the gene exhibit the extreme phenotype (*Figure 1D*; *Supplementary file 1B*).

Next, we used cell-specific molecular markers to characterize the most affected *Sox9*-deficient retinas. Initially, we focused on the MG cell layer by means of SOX8 immunofluorescence. We counted the number of SOX8+ cells per 100 µm of INL, and we found a significant reduction in the number of MG cells in mutant retinas (control, 18.5 ± 3.5, *n* = 7; mutant, 10.5 ± 1.2, *n* = 6; p = 0.000107, Mann–Whitney *U*-test; *Figure 1E*; *Supplementary file 1C*). In addition to the reduced number of SOX8+ cells, we observed areas within some *Sox9*-mutant INL layers that completely lacked SOX8+ cells (*Figure 1C*, arrowheads), indicating that these regions were entirely devoid of MG cells.

To independently validate the loss of MG cells in *Sox9*-deficient retinas, we examined the expression of S100, a cytoplasmic marker that labels the processes of adult Müller cells. In control retinas,

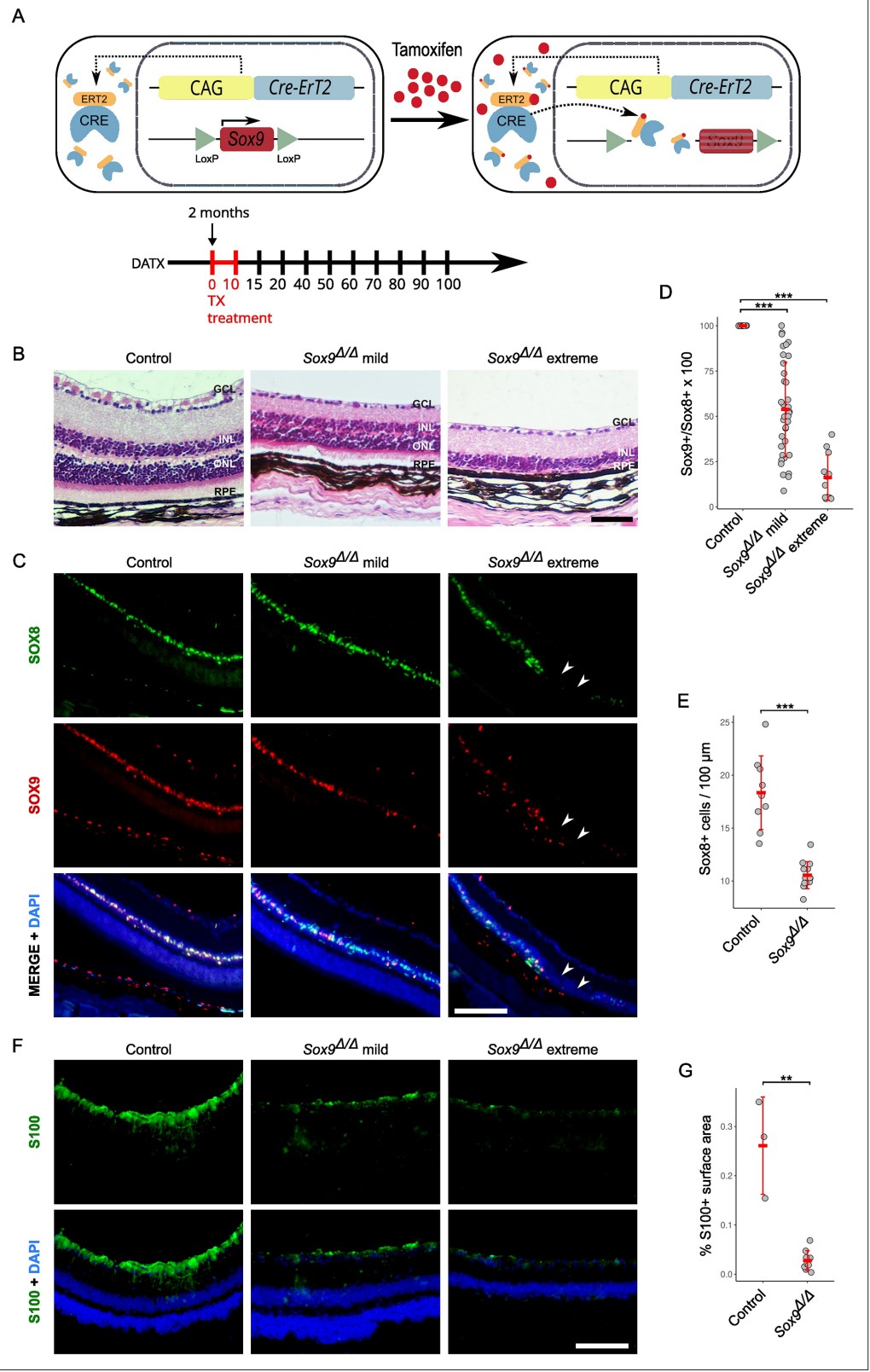

**Figure 1.** Efficiency and morphological impact of *Sox9* deletion on the adult mouse retina. (**A**) Schematic of the experimental design. Adult *Sox9^flox/flox^; CAGG-CreER* mice were treated with tamoxifen (TX) at 2 months of age to induce ubiquitous *Sox9* deletion. Retinal samples were collected at various days after tamoxifen treatment (DATX). (**B**) Hematoxylin–eosin stained histological sections of retinas from control and *Sox9^Δ/Δ^* mice. Mutant retinas

*Figure 1 continued on next page*

*Figure 1 continued*

exhibit variability in phenotype severity, with some showing mild morphological defects and a few displaying an extreme phenotype characterized by the loss of the outer nuclear layer (ONL). (**C**) Double immunofluorescence for SOX8 (green) and SOX9 (red) in retinal sections from adult control and *Sox9*$^{\Delta/\Delta}$ mice. The severity of phenotypes correlates with the extent of *Sox9* inactivation. Regions completely lacking SOX8$^+$ cells are indicated by arrowheads. DAPI (blue) was used to counterstain nuclei. (**D**) Quantification of the percentage of SOX8$^+$ cells co-expressing SOX9 in control and *Sox9*$^{\Delta/\Delta}$ retinas with mild and extreme phenotypes. (**E**) Quantification of SOX8$^+$ cells per 100 μm of inner nuclear layer (INL) length in control and *Sox9*$^{\Delta/\Delta}$ retinas. (**F**) Immunofluorescence for S100 showing strong labeling of Müller cell processes in control retinas and a progressive reduction in *Sox9*$^{\Delta/\Delta}$ samples. (**G**) Quantification of S100+ signal in control and *Sox9*$^{\Delta/\Delta}$ mice expressed as percentage of surface area occupied in the retina. The black scale bar represents 50 μm in **B**, and the white scale bars represent 100 μm in **C** and 50 μm in **F**. GCL, ganglion cell layer; INL, inner nuclear layer; RPE, retinal pigmented epithelium. Mann–Whitney *U*-test, p < 0.01 (**\*\***), p < 0.001 (**\*\*\***).

strong S100 immunoreactivity was observed across the inner retina, outlining the typical radial projections of Müller glia (*Figure 1G*). In contrast, *Sox9*$^{\Delta/\Delta}$ retinas with an extreme phenotype exhibited a marked reduction in S100 signal (*Figure 1F*). Given the diffuse cytoplasmic localization of S100, we quantified its expression by measuring the fluorescence signal within a defined surface area of the retina. This analysis revealed a statistically significant reduction in S100 signal intensity in mutant samples (including both mild and extreme phenotypes) compared to controls (*Figure 1G*; *Supplementary file 1D*), further supporting the loss of MG cells upon *Sox9* deletion.

Since the main observable consequence of *Sox9* deletion in the retina of adult mice showing the extreme phenotype is the absence of the ONL layer, which harbors both rod and cone photoreceptors, we further examined the status of these cell types by immunofluorescence. Cone photoreceptor cells were assessed by retinal whole-mount staining with two different opsins: OPN1SW (short-wavelength-sensitive opsin, referred to as S opsin) and OPN1LW (medium- and long-wavelength-sensitive opsin, referred to as M opsin). In control retinas, we observed uniform staining on the ventral surface and on the entire surface for S opsin and M opsin, respectively. In contrast, in *Sox9*-deficient retinas, we could only identify small isolated immunofluorescent patches (*Figure 2A*, upper). Double immunofluorescence on retinal sections confirmed that mutant retinas lacked expression for both cone photoreceptors (*Figure 2A*, bottom). We also performed immunofluorescence for RHO, a marker for rod photoreceptor cells, on retinal whole mounts and sections. We found a pattern of expression similar to that described for M opsin in both the control and mutant retinas. These results evidence that *Sox9* is required for the maintenance of the two photoreceptor cell types present in the adult mouse retina (*Figure 2B*).

We also studied the status of other retinal cell types. The transcription factor BRN3A was used to identify ganglion cells (*Nadal-Nicolás et al., 2009*), which were shown to decrease in number in the mutant retinas, compared to control ones (*Figure 2C, D* and *Supplementary file 1E*; *n* = 5 controls, *n* = 12 mutants; Mann–Whitney *U*-test, p = 3 × 10⁻⁴). Similarly, double immunodetection of the transcription factors PAX6 and AP2A was used to identify both amacrine and horizontal cells (*Figure 2E*), as previously described (*Marquardt et al., 2001*; *Barnstable et al., 1985*; *Edqvist and Hallböök, 2004*), showing a similar reduction in both cell types in degenerated retinas (*Figure 2F, G* and *Supplementary file 1F*; AP2$\alpha$+ amacrine cells: *n* = 3 controls, *n* = 8 mutants; two-sample *T*-tests, p = 0.029; PAX6+/AP2α– horizontal cells: *n* = 3 controls, *n* = 8 mutants; Mann–Whitney *U*-test, p = 0.021). These findings indicate that the loss of *Sox9* in the adult retina ultimately leads to the degeneration of multiple inner retinal neuronal populations, beyond the previously described effects on photoreceptors and Müller glia.

## Photoreceptors undergo apoptosis in the absence of *Sox9*

Since photoreceptors are absent in severely affected *Sox9*-mutant retinas, we conducted TUNEL assays to study the role of cell death in the process of retinal degeneration. In control samples (*n* = 5), almost no TUNEL signal was observed in the retina. In contrast, *Sox9*$^{\Delta/\Delta}$ mice (*n* = 15) showed numerous TUNEL+ cells, mainly located in the persisting ONL, indicating that photoreceptor cells were dying (*Figure 3A*). Although extensive TUNEL staining in the ONL was clearly observed in two Sox9$^{\Delta/\Delta}$ retinas with mild phenotypes, this pattern was not consistently present across the full cohort. In the remaining 13 mutant retinas, we observed a modest but noticeable increase in the number of

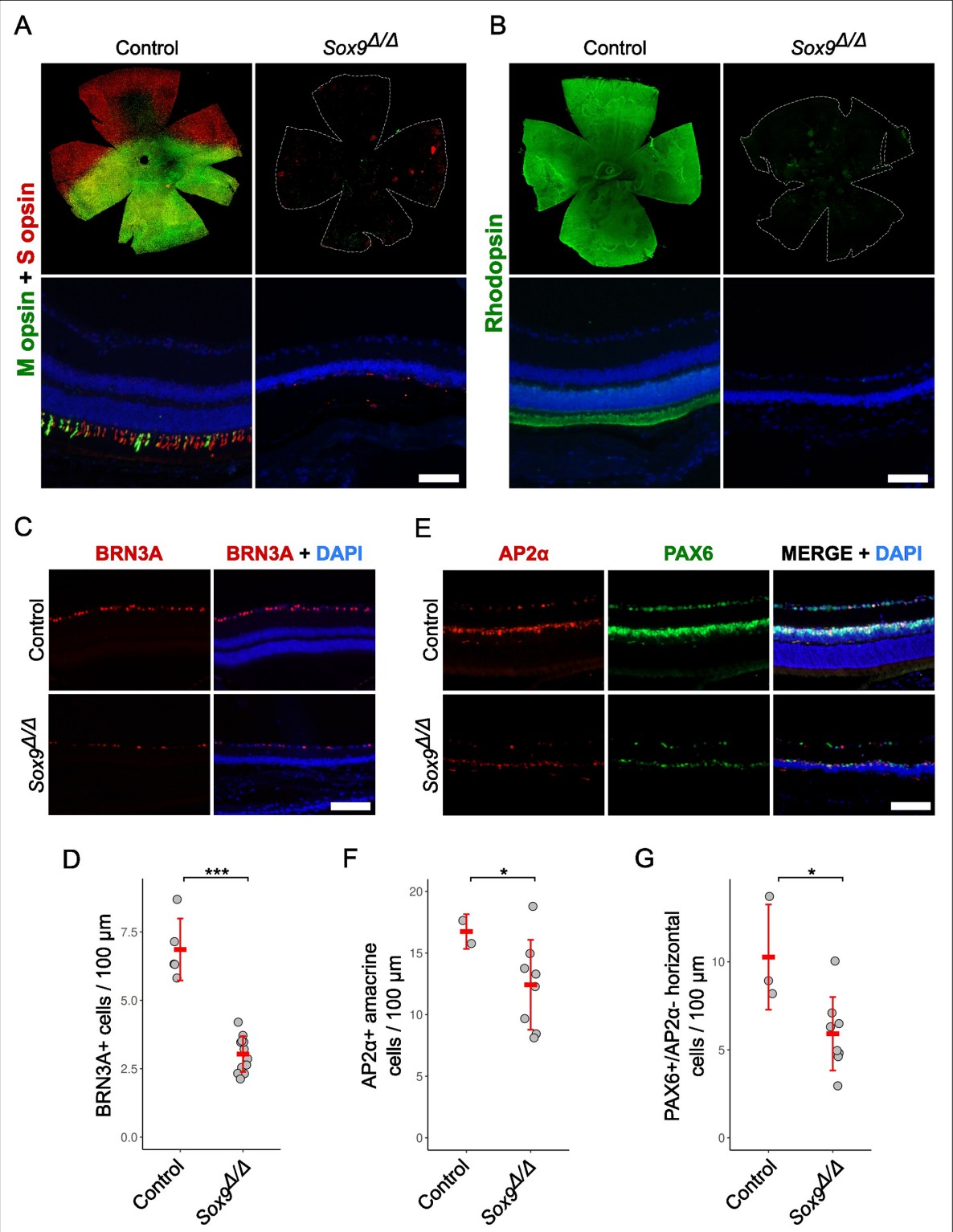

**Figure 2.** Immunodetection of retinal cell markers in adult retinas from control and *Sox9*[Δ/Δ] mice. (**A**) Immunofluorescence analysis of cone photoreceptor cells in retinal whole mounts (upper images) and histological sections (lower images) using double staining for OPN1SW (S opsin) and OPN1LW (M opsin). (**B**) Immunofluorescence analysis of rod photoreceptor cells in retinal whole mounts (upper images) and histological sections (lower images) using Rhodopsin staining. (**C**) Immunofluorescence analysis of ganglion cells in retinal sections using BRN3A staining. (**D**) Quantification of BRN3A⁺ cells per 100 μm of retinal ganglion cell nuclear layer length in control and *Sox9*[Δ/Δ] retinas. (**E**) Immunofluorescence analysis of amacrine and horizontal cells in retinal sections using double staining for PAX6 and AP2α. (**F**) Quantification of AP2α+ amacrine cells in control and *Sox9*[Δ/Δ] mice per

*Figure 2 continued on next page*

*Figure 2 continued*

100 µm of inner nuclear layer. (**G**) Quantification of PAX6+/ AP2α− horizontal cells in control and *Sox9*^Δ/Δ mice per 100 µm of inner nuclear layer. DAPI (blue) was used to counterstain nuclei. The scale bars in **A** and **B** represent 1 mm for the top row and 50 µm for the bottom row; the scale bars in **C** and **E** represent 90 µm. Mann–Whitney *U* and two-sample *T*-tests, p < 0.05 (*), p < 0.001 (***).

apoptotic cells compared to controls (*Figure 3B*; *Supplementary file 1G*). Despite a high frequency of zero counts (particularly among controls), the difference between groups reached statistical significance when analyzed using a zero-inflated Poisson model (p = 0.028; *n* = 5 controls, 13 mutants). These findings suggest that photoreceptor apoptosis following Sox9 deletion may occur acutely and within a narrow temporal window, making it challenging to capture the full degenerative process at a single time point.

## MG cells undergo reactive gliosis in *Sox9*-deficient retinas

Finally, we decided to study the expression of the glial fibrillary acidic protein (GFAP), whose expression is upregulated in MG cells undergoing reactive gliosis associated with retinal cell injury (*Ekström et al., 1988*). As previously reported (*Fernández-Sánchez et al., 2015*), in control mice, GFAP immunostaining was mainly limited to the ganglion cell layer, in the inner margin of the retina. However, in *Sox9*-deficient retinas with a mild phenotype, GFAP-positive processes reached the INL layer. In mutant retinas with an extreme phenotype, GFAP-positive MG processes were distributed throughout the entire thickness of the degenerated retinas. Thus, in the absence of *Sox9*, MG cells seem to undergo a progressive process of gliosis (*Figure 3C*). To support these observations quantitatively, we

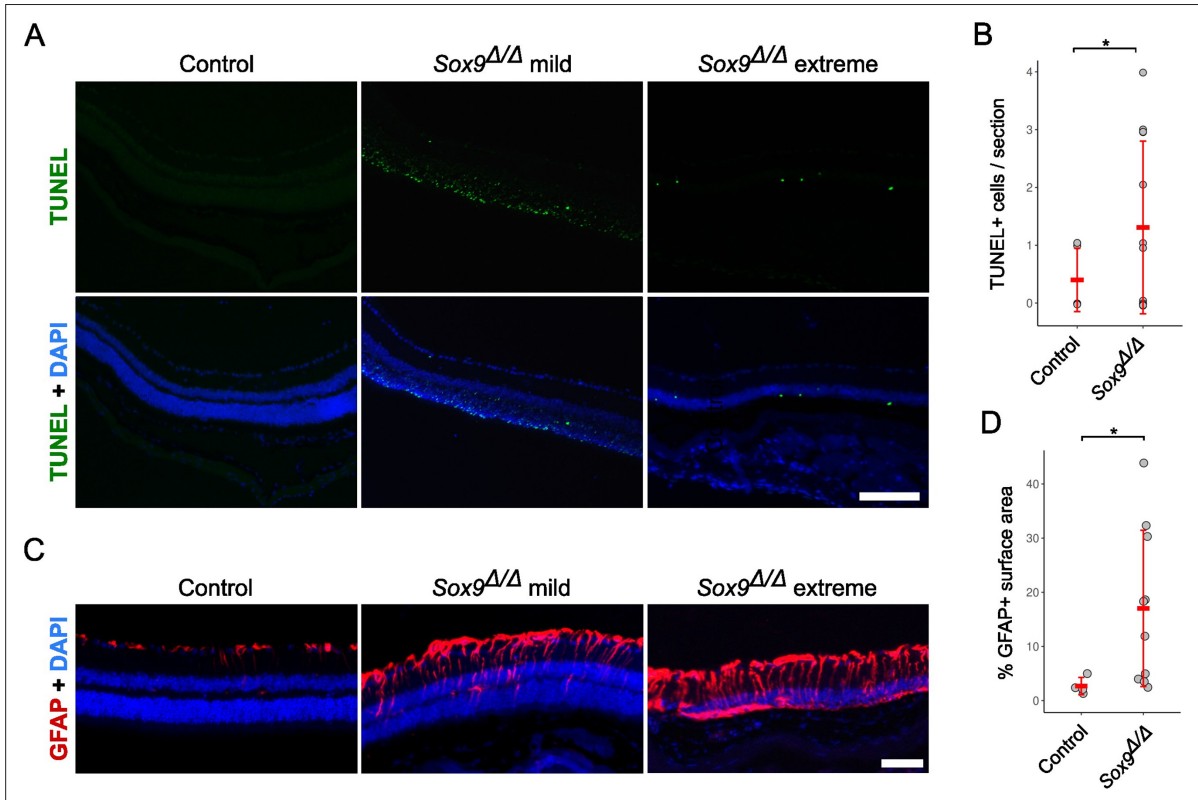

**Figure 3.** Assessment of retinal damage and apoptosis in *Sox9*-deficient mice. (**A**) TUNEL staining of adult retinas from control and *Sox9*^Δ/Δ mice with mild and extreme phenotypes. A large number of TUNEL-positive photoreceptor cells is evident in two mutant mice with mild phenotypes, indicating extensive apoptotic events affecting the outer nuclear layer. (**B**) Quantification of the total number of TUNEL+ cells among control and *Sox9*^Δ/Δ mice without extensive apoptosis in 20x microphotographs of retinal sections stained for TUNEL. (**C**) Immunofluorescence staining of GFAP in adult retinas from control and *Sox9*^Δ/Δ mice with mild and extreme phenotypes. Müller glial cell activation is observed in *Sox9*-deficient retinas, with GFAP expression extending across the entire thickness of the retina in extreme phenotypes, suggesting progressive gliosis. (**D**) Quantification of GFAP+ signal in control and *Sox9*^Δ/Δ mice expressed as percentage of surface area occupied in the retina. Mann–Whitney *U*-test (**B**) and zero-inflated Poisson test (**D**), p < 0.05 (*). The scale bars represent 50 µm.

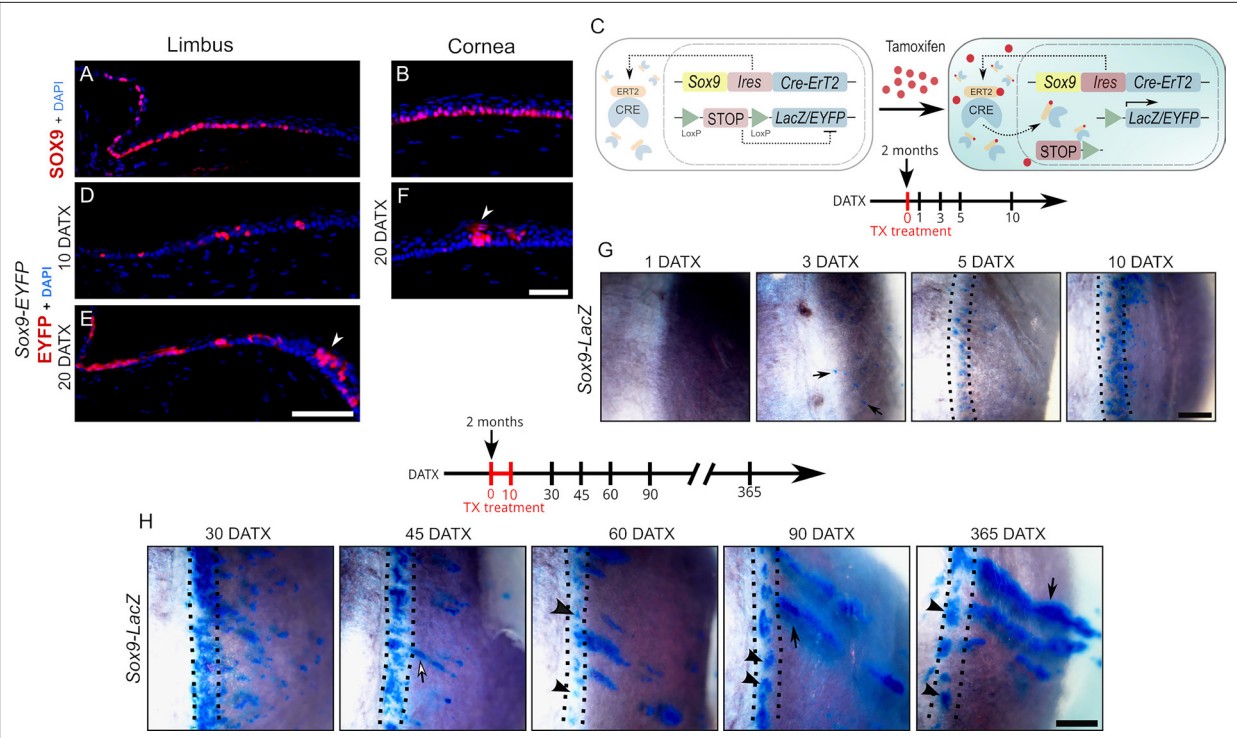

**Figure 4.** Analysis of the fate of SOX9-expressing cells through lineage tracing experiments. Immunofluorescence for SOX9 in limbal (**A**) and corneal (**B**) sections of adult control mice. DAPI (blue) was used to counterstain nuclei. (**C**) Schematic of the genetic lineage tracing strategy. Tamoxifen (TX) was administered to label SOX9-expressing cells and their progeny. Analysis of EYFP expression in the limbus (**D, E**) and cornea (**F**) of *Sox9*-EYFP adult mice. At 10 days after TX administration (10 DATX), discrete EYFP-positive clones are visible in the limbal region (**D**). By 20 DATX, these clones expand toward the peripheral cornea, with some cells reaching the outer epithelial layers at the limbus–cornea border (arrowhead in **E**). In the central cornea, EYFP-positive cells extend along the entire epithelium by 20 DATX (arrowhead in **F**). (**G**) Short-term whole-mount X-gal staining of *Sox9*-LacZ eyes. The first LacZ-positive cells scattered across the limbal and corneal surface appeared at 3 DATX (arrows). (**H**) Long-term whole-mount X-gal staining of *Sox9*-LacZ eyes. At 45 DATX, LacZ-positive stripes emerge from the limbus and extend into the peripheral cornea (arrow). At 60 DATX, circumferential clones are observed in the limbus (arrowheads). At 90 DATX, three clone types are visible: circumferential clones in the limbus (arrowheads), stripes from the limbus reaching the central cornea (arrow), and stripes without a base in the limbus (asterisk). At 365 DATX, the first two clone types are mainly observed, though in reduced numbers and larger sizes (see arrowheads and arrow as in the 90 DATX picture), with some stripes without a base in the limbus (asterisk). Dashed lines in **G** and **H** enclose the limbal area. Scale bars in **E** and **F** represent 50 µm; scale bars in **G** and **H** represent 500 µm.

The online version of this article includes the following figure supplement(s) for figure 4:

**Figure supplement 1.** Whole-mount X-gal staining of control and Sox9-LacZ eyes at different days after tamoxifen (DATX) administration.

Scale bar represents 1 mm.

measured GFAP fluorescence intensity across defined retinal surface areas in control and $Sox9^{\Delta/\Delta}$ mice (*Figure 3D*; *Supplementary file 1H*). This analysis revealed a statistically significant increase in GFAP signal in mutant retinas compared to controls (Mann–Whitney $U$-test, p = 0.024; $n$ = 4 controls, 10 mutants). These results are consistent with a progressive gliotic response following *Sox9* deletion and provide further evidence that MG cells become reactive in the absence of *Sox9*.

## Lineage tracing of *Sox9*-expressing cells in the limbus and the cornea

We next decided to shed light on the role of *Sox9* in the adult cornea. Previous studies have documented the expression of *Sox9* in the adult mouse limbal and central cornea epithelium (*Menzel-Severing et al., 2018*; *Sartaj et al., 2017*). In line with these findings, we also observed the presence of the SOX9 protein in the basal cells of both structures (*Figure 4A, B*), which are the sites housing corneal progenitor and stem cells. To assess the stemness potential of these *Sox9*-positive cells, we conducted TX-inducible genetic lineage tracing by breeding mice harboring an IRES-CreER[T2] cassette inserted into the endogenous *Sox9* locus, *Sox9*[IRES-CreERT2] (*Soeda et al., 2010*), to either *Rosa26*-EYFP (*Srinivas et al., 2001*) or *Rosa26-LacZ* (*Soriano, 1999*) reporter mice. We induced labeling of

*Sox9*-expressing cells and their progeny by administering TX to adult mice (2 months old) and analyzed reporter expression at different DATX (**Figure 4C**). No X-gal- or EYFP-positive cells were detected at any of the stages analyzed in the absence of TX (**Figure 4—figure supplement 1**, control panels). Immunofluorescence for EYFP on limbal *Sox9IRES-CreERT2- Rosa26-EYFP (Sox9-EYFP)* sections at 10 DATX unveiled the presence of discrete clones of EYFP-positive cells (**Figure 4D**). By 20 DATX, these clones were larger and extended toward the peripheral cornea. At this latter stage, EYFP-positive clones with cells reaching the outer layers of the epithelium were visible at the limbus cornea border (**Figure 4E**, arrowhead). At 20 DATX, we also observed groups of cells spanning the entire apical-basal axis of the central corneal epithelium (**Figure 4F**, arrowhead). These results demonstrate that SOX9⁺ cells residing in the basal layer of the limbal and corneal epithelium are progenitors of the terminally differentiated cells of the corneal epithelium.

We then investigated the dynamics of stem cell progression through whole-mount X-gal staining of *Sox9*ᴵᴿᴱˢ⁻ᶜʳᵉᴱᴿᵀ²;*Rosa26-LacZ (Sox9-LacZ)* eyes. At first, we conducted a short-term study with a single TX injection and analyzed X-gal staining for up to 10 days after treatment (**Figure 4G** and **Figure 4—figure supplement 1**). We observed the first X-gal-positive cells at 3 DATX, and they were scattered throughout the entire limbal and corneal surface (**Figure 4G**, arrows). At 5 DATX, most X-gal⁺ cells were located in the limbal area, and by 10 DATX, the number of stained cells increased notably in this area. These observations suggest that *Sox9*-expressing descendant cells experience a process of proliferation in the limbus, illustrating their transient cell amplification behavior. To gain insights on the long-term cloning capacity and contribution dynamics of *Sox9*-progenitor cells, we administered tamoxifen for 10 days in the food and performed X-gal staining from 30 DATX to 365 DATX (**Figure 4H** and **Figure 4—figure supplement 1**). At 30 DATX, LacZ was mostly expressed in the limbus, although lacZ⁺ clones could also be observed throughout the peripheral and central cornea. Overall, these clones were larger than those observed at 10 DATX. Fifteen days later (45 DATX), the limbal staining began to disperse, allowing the observation of discrete clones. Simultaneously, LacZ⁺ stripes emerging from the limbus and extending into the peripheral cornea became apparent (**Figure 4H**, 45 DATX, arrow, and **Figure 4—figure supplement 1**). This situation was evident at 60 DATX, when the thickness and number of the stripes had increased. Additionally, starting at this stage, circumferential clones that remained in the limbus could also be appreciated (**Figure 4H**, 60 DATX, arrowheads, and **Figure 4—figure supplement 1**). At 90 DATX, three types of clones were observed: circumferential ones in the limbus (**Figure 4H**, 90 DATX, arrowheads, and **Figure 4—figure supplement 1**), stripes originating from the limbus, which often reached the central cornea (**Figure 4H**, 90 DATX, arrow, and **Figure 4—figure supplement 1**), and stripes without a base in the limbus (**Figure 4H**, 90 DATX, asterisk, and **Figure 4—figure supplement 1**). At 365 DATX, the first two types of clones were mainly observed, albeit in reduced numbers and larger sizes. We also observed some stripes without a base in the limbus (**Figure 4H**, 365 DATX, asterisk, and **Figure 4—figure supplement 1**). Thus, *Sox9* is expressed in a basal limbal stem cell population with the ability to form two types of long-lived cell clones involved in stem cell maintenance and homeostasis.

## *Sox9* is involved in limbal stem cell differentiation

To gain insight into the function of *Sox9* as a limbal stem cell marker, we used the single-cell RNA-sequencing dataset of isolated epithelial cells from the limbus (with marginal conjunctiva and corneal periphery) recently generated by **Altshuler et al., 2021**. In this study, the authors describe the co-existence of two separated stem populations in the limbus, the quiescent 'outer' limbus (OLB) one, adjacent to the conjunctiva, and the active 'inner' limbus (ILB) one, adjacent to the peripheral cornea. Unbiased clustering of the scRNA-seq dataset and subsequent assignment of cell identity revealed the existence of 10 cell populations equivalent to those identified by Altshuler et al. (**Figure 5A** and **Figure 5—figure supplement 1A**). Density plots based on gene-weighted kernel density estimation and violin plots of expression levels revealed high expression levels of *Sox9* in OLB cells, and progressively weaker expression in the ILB, corneal basal, and corneal suprabasal clusters (**Figure 5B, C**). We also detected strong expression for *Sox9* in a mitotic cluster (**Figure 5B**). Next, we inferred the trajectory of limbal and corneal cells by using the partition-based graph abstraction (PAGA) algorithm and reconstructing the temporal order of differentiating cells with the diffusion pseudotime software, using the OLB cluster as 'root' (**Wolf et al., 2019**; **Haghverdi et al., 2016**). The PAGA graph revealed a strong connection between the OLB node and a mitotic cell population with high expression of

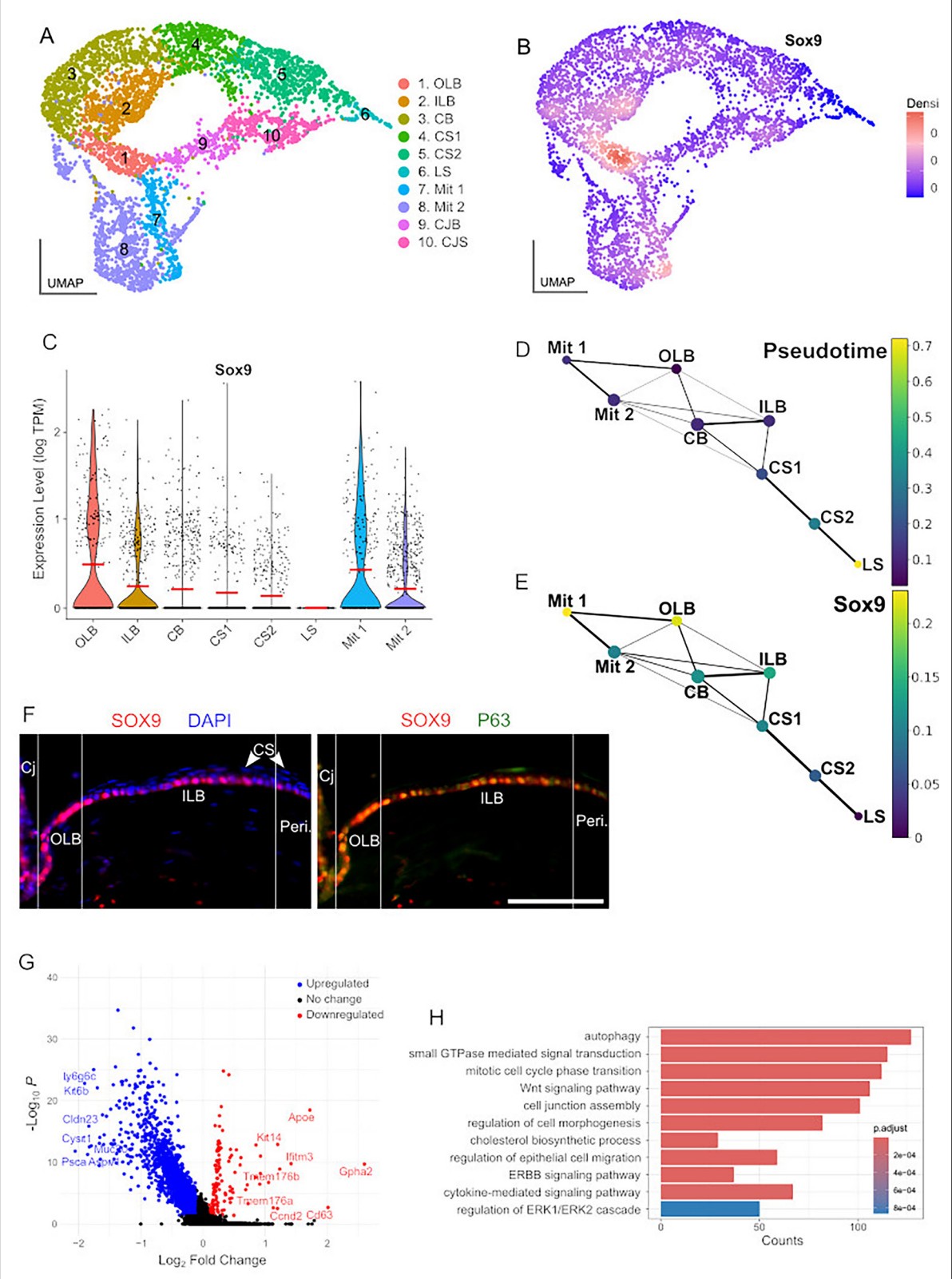

**Figure 5.** Single-cell RNA-sequencing (scRNA-seq) analysis of *Sox9* expression in the mouse limbal and corneal epithelium. (**A**) Uniform manifold approximation and projection (UMAP) visualization of scRNA-seq data of isolated limbal epithelial cells from *Altshuler et al., 2021*. (**B**) Density plot of *Sox9* expression. (**C**) Violin plot of *Sox9* expression levels across the different cell clusters identified in **A**. (**D**) Pseudotime partition-based graph abstraction (PAGA) graph of the trajectory of limbal and corneal cells. (**E**) *Sox9* expression in the trajectory of cell differentiation described in **D**.

*Figure 5 continued on next page*

Figure 5 continued

(F) Immunofluorescence for SOX9 (left) and SOX9 and P63 (right) in the mouse limbal region. (G) Volcano plot of differentially expressed genes (DEG) between cells exhibiting high and low *Sox9* expression levels. (H) Gene ontology analysis of DEG identified in G. CB, corneal basal; Cj, conjunctiva; CJB, conjunctiva basal; CJS, conjunctiva suprabasal; CS1, corneal suprabasal 1; CS2, corneal suprabasal 2; ILB, inner limbal basal; LS, limbal suprabasal; Mit 1, mitotic 1; Mit 2, mitotic 2; OLB, outer limbal basal; Peri, pericorneal region.

The online version of this article includes the following figure supplement(s) for figure 5:

**Figure supplement 1.** Dot plot showing the expression of specific markers in the cell groups identified from the single-cell RNA-seq dataset of isolated limbal epithelial cells reported by *Altshuler et al., 2021*.

OLB and stem cell markers (*Gpha2*, *Ifitm3*, *Cd63*, and *Krt15*), which, in turn, was strongly linked to a second mitotic cell cluster, in which the expression of these markers diminished while the expression of corneal markers (*Krt12*, *Ppp1r3c*, and *Slurp*) increased (*Figure 5D*, *Figure 5—figure supplement 1B*). Through the PAGA graph, we also observed that the expression levels of *Sox9* correlated with the pseudotime graph (*Figure 5E*), indicating that *Sox9* expression decreases as transiently amplifying progenitors undergo progressive differentiation from limbal to peripheral corneal cells. To validate these results, we decided to closely examine *Sox9* expression in the limbus using immunofluorescence. Previous analyses revealed that the outer limbus is approximately 100 µm wide, while the inner limbus is wider, around 240 µm (*Altshuler et al., 2021*). We observed that in the region corresponding to the OLB, most cells showed strong *Sox9* expression. In the area corresponding to the ILB, this immunoreactivity appeared weaker in the basal layer (corresponding to the ILB proper), and no expression was detected in the suprabasal layers (flattened cells; *Figure 5F*, left). Double immunofluorescence for SOX9 and P63, which is expressed in basal cells of the limbal epithelium, but not by transient amplifying cells covering the corneal surface (*Pellegrini et al., 2001*) revealed that Sox9 expression was restricted to P63-positive cells (*Figure 5F*, right). These observations confirm that Sox9 is expressed in a basal cell population within both the OLB and ILB and that its expression decreases in differentiated transient amplifying cells. Next, we conducted a comparative analysis of gene expression to examine the differences between cells exhibiting high (≥0.7) and low expression levels (between 0.7 and 0.15) of Sox9. We identified 2816 deregulated genes, 137 upregulated, and 2679 downregulated (adjusted p-value <0.05; $|avg\_log_2FC| > 0.1$; *Figure 5G*; *Supplementary file 1I*). Among the upregulated genes, we found OLB markers including Gpha2, Ifitm3, and Cd63, in addition to stem cell markers such as Krt14 (*Figure 5G*; *Supplementary file 1I*).

Finally, we performed gene ontology analysis using the downregulated genes, and we found terms related to stem cell differentiation, such as 'mitotic cell phase transition' (*Knoblich, 2008*), 'autophagy' (*Chang, 2020*), 'cell–cell junction' (*Ning et al., 2021*), 'regulation of epithelial cell migration' (*Puri et al., 2020*), 'regulation of cell morphogenesis' and 'stem cell differentiation', as well as gene pathways involved in these processes including 'Wnt-signaling pathway', 'ERBB signaling' (*Hassan and Seno, 2022*), 'cytokine-mediated signaling pathway' (*Korkaya et al., 2011*), and 'regulation of ERK1 and ERK2 cascade' (*Lavoie et al., 2020*; *Figure 5H*; *Supplementary file 1J*). Altogether, these results suggest that the population of SOX9-negative cells is more differentiated than that of SOX9-positive cells.

### *Sox9*$^{\Delta/\Delta}$ progenitor cells lose their clonogenic capacity

Our results suggest that Sox9 is essential for limbal stem cell differentiation. To test this hypothesis, we used the same tamoxifen-inducible conditionally deleted Sox9 gene in adult mouse cells (2 months old) that we used in the retina study. However, histological examination of the limbus and cornea from Sox9Δ/Δ mice at any time up to and including 100 DATX did not reveal any discernible phenotypic effects (*Figure 6A*). Immunofluorescence staining for SOX9 (*Figure 6B*) failed to reveal any differences in the number of SOX9-expressing cells between mutants and controls (*Figure 6C*; *Supplementary file 1K*; control, 5.4 ± 0.4 cells/100 µm, *n* = 7; mutant, 5.9 ± 0.7 cells/100 µm, *n* = 6; p = 0.147, Welch *t* test). Similar observations were made for corneal and limbal marker genes such as *P63* (*Figure 6B, C*; *Supplementary file 1K*; control, 6.1 ± 0.7 cells/100 µm, *n* = 6; mutant, 6.1 ± 0.7 cells/100 µm, *n* = 5; p = 0.147, Welch *t* test) and *Pax6* (*Figure 6—figure supplement 1A*). These observations are consistent with a model whereby tamoxifen induces mosaic patterns of cell *Sox9*-deleting in the ocular surface, but that the *Sox9*-null cells cannot survive or proliferate as well as their wild-type neighbors and are hence outcompeted over time, leading to an essentially wild-type cornea. We tested this model by

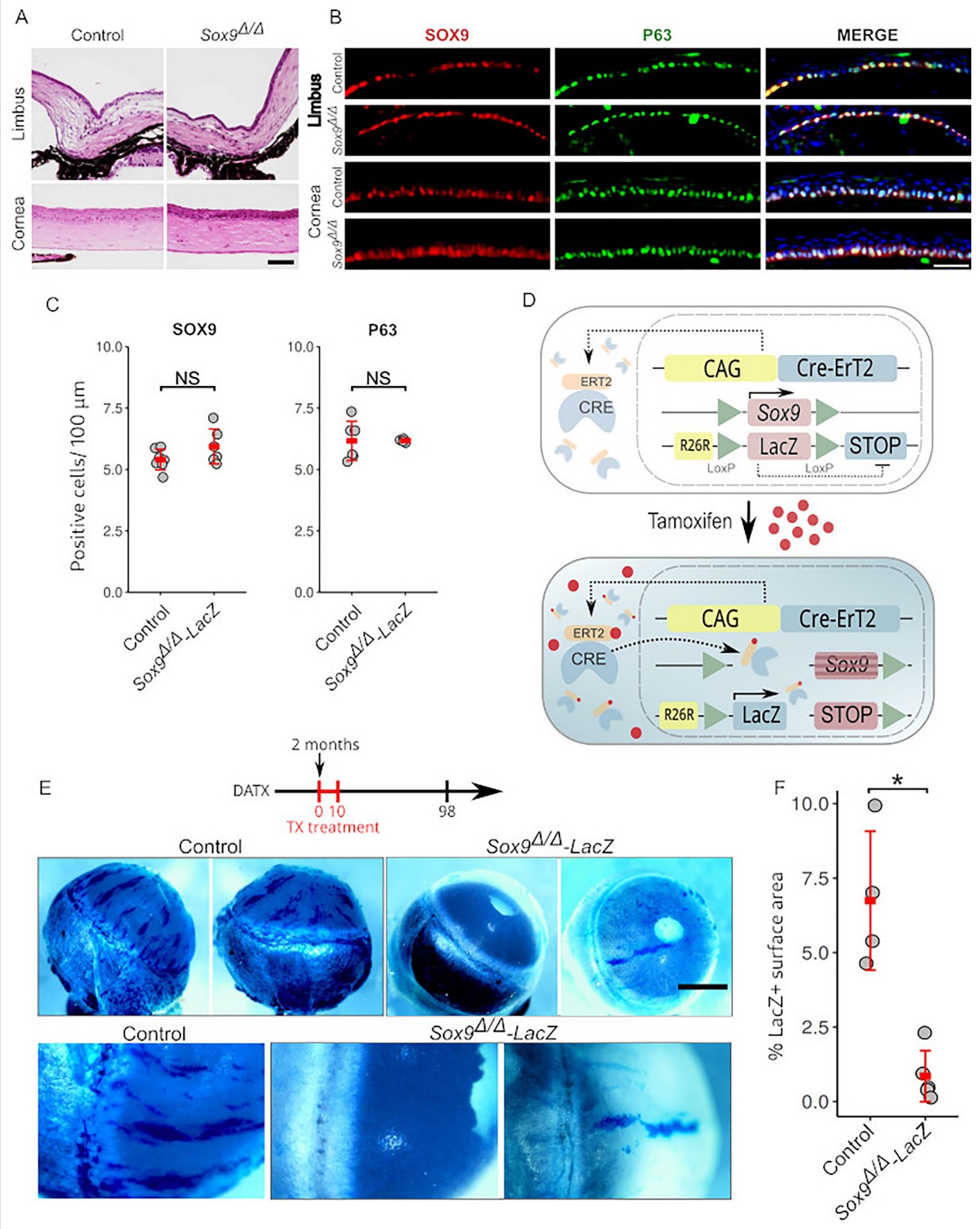

**Figure 6.** Impact of *Sox9* deletion on limbal stem cell differentiation and clonogenic capacity. (**A**) Hematoxylin–eosin stained histological sections of the limbus and cornea from control and *Sox9^{Δ/Δ}* mice. (**B**) Immunofluorescence staining for SOX9 (red) and P63 (green) in the limbus and cornea from control and *Sox9^{Δ/Δ}* mice. (**C**) Quantification of SOX9+ and P63+ cells per 100 µm of corneal limbus section length in control and *Sox9^{Δ/Δ}* mice. (**D**) Schematic of the generation of *Sox9^{Δ/Δ}*-LacZ mice. Tamoxifen (TX) was administered for 10 days to label *Sox9*-deleted cells and their progeny. (**E**) X-gal staining of whole eyes from control and *Sox9^{Δ/Δ}*-LacZ mice at 98 days after TX administration. Control corneas exhibited numerous LacZ+ patches and elongated

*Figure 6 continued on next page*

*Figure 6 continued*

stripes originating from the limbus. In contrast, minimal X-gal staining was observed in $Sox9^{\Delta/\Delta}$-LacZ corneas, indicating impaired clonogenic capacity in $Sox9$-deleted cells. (**F**) Quantification of X-gal-stained surface areas in control and $Sox9^{\Delta/\Delta}$-LacZ mice. Each data point represents an individual mouse. Mann-Whitney $U$-test, p < 0.05 (*). Scale bar in **A** represents 100 µm; scale bar in **B** represents 50 µm; scale bar in **D** represents 1 mm for the top row and 500 µm for the bottom row.

The online version of this article includes the following figure supplement(s) for figure 6:

**Figure supplement 1.** Effect of *Sox9* deletion on PAX6 expression and stem cell differentiation in the limbus and cornea.

leveraging this mosaicism to investigate the in vivo clonal capacity of both *Sox9*-positive and *Sox9*-negative cells. For this, we generated *CAGG-CreERTM;Sox9^{flox/flox};Rosa26-LacZ* mice ($Sox9^{\Delta/\Delta}$-LacZ), in which *Sox9*-deleted cells express the *LacZ* gene, and therefore, are labeled in blue after X-gal staining (*Figure 6D*). As controls, we used *CAGG-CreERTM;Sox9^{flox/+};Rosa26-LacZ* mice ($Sox9^{\Delta/+}$-LacZ). Adult mice (2 months old) were fed with a tamoxifen-supplemented diet for 10 days to induce *LacZ* expression followed by a chase period of 14 weeks. This time frame aligns with previous observations reporting the appearance of clear X-gal$^+$ stripes in *CAGG-CreERTM;Rosa26-LacZ* mice (*Dorà et al., 2015*; *Dorà et al., 2015*). In control $Sox9^{\Delta/+}$-LacZ corneas, derived from four mice we could observe numerous discrete X-gal$^+$ patches of different sizes spanning over the entire corneal surface. In all cases, we observed a variable number of elongated stripes originating from the limbus, suggesting that they come from limbal stem cells (*Figure 6E* and *Figure 6—figure supplement 1B*). In contrast, the majority of $Sox9^{\Delta/\Delta}$-LacZ corneas, derived from five mice exhibited minimal blue staining, with only occasional, small X-gal$^+$ spots observed. Notably, only one mutant cornea revealed the presence of an elongated stripe originating from the limbus (*Figure 6E*, right panels; *Figure 6—figure supplement 1B*). A possible explanation for this clone may be that spontaneous ligand-independent activity of Cre-ER fusion may have occurred in a bona fide limbal stem cell, as previously reported (*Vooijs et al., 2001*; *Kemp et al., 1994*; *Haldar et al., 2008*; *Dorà et al., 2015*). To quantify this difference, we measured the percentage of blue-stained surface area, using the animal as the biological replicate. For control animals where both eyes were analyzed, the average value was used. This analysis confirmed a significant decrease in the mutant corneas compared to the controls (controls: n = 4 mice $Sox9^{\Delta/+}$-LacZ, 6.74 ± 2.33; mutants: n = 5 mice $Sox9^{\Delta/\Delta}$-LacZ, 0.85 ± 0.86; Mann–Whitney $U$-test, p = 0.0159; *Figure 6F*; *Supplementary file 1L*). These results reveal that during normal in vivo homeostasis, epithelial limbal and corneal stem/progenitor cells require *Sox9* to maintain their clonogenic capacity, indicating a key role for this gene in the process.

## Discussion

*SOX9* is a master regulatory gene involved in the specification and/or development of embryonic tissues belonging to the three germ layers. In recent years, several studies revealed that its expression is maintained in adult ectoderm and endoderm-derived cells, indicating its role in the homeostasis of mature organs (*Jo et al., 2014*). Here, we analyze the in vivo role of the gene in two structures belonging to the adult mouse eye, the retina and cornea, with completely different anatomy and biological functions. We conditionally deleted *Sox9* in the adult mouse retina and observed a phenotype characterized by different degrees of retinal degeneration. In the most affected individuals, this degeneration leads to the disappearance of the ONL layer and almost complete absence of cone and rod photoreceptors. Since *Sox9* is expressed in MG and RPE cells (*Poché et al., 2008*), it is challenging to determine the contribution of each cell type to the final mutant phenotype. Previous studies on *Sox9*-mutant mice have not described any eye phenotype similar to the one we report here. The main reason is probably that these studies focused on the developing retina and are therefore not useful for understanding how the absence of *Sox9* in MG and RPE cells in the adult eye leads to the retinal degeneration observed in our *Sox9*-deficient adult mutants. In this context, we can obtain better insights by examining the physiological response to RPE and MG cell loss after Cre-mediated activation of diphtheria toxin A (DTA) in these cell types. In the first case, the major retinal abnormality was the formation of regions of photoreceptor rosetting with very mild degeneration of the photoreceptors (*Longbottom et al., 2009*). In mice with a DTA-induced ablation of MG cells, a wave of cell apoptosis was observed in the ONL but not in the other retinal layers (*Shen et al., 2012*). In our mutant mice, we did not observe photoreceptor rosetting in the ONL, but a massive apoptosis

in this retinal layer. In addition, we observed that some regions within the INL layer were devoid of MG cells. Thus, we favor the hypothesis that continuous expression of *Sox9* in MG cells is necessary for normal function and/or survival of this cell type. In the absence of *Sox9*, MG cells undergo massive retinal gliosis, and they cannot provide structural, metabolic, and functional support to the other retinal cells, leading to a massive loss of photoreceptors followed by the depletion of other retinal cell types. In agreement with this, *Sox9* expression is upregulated in MG cells after retinal light damage in rats (*Wang et al., 2013*). However, the retinal phenotype we report here is much more severe than the one observed in both DTA-induced transgenic lines. This could be due to the different efficiency of Cre-induced recombination, but also to the fact that the combined deletion of *Sox9* in both MG and RPE may cause a more severe phenotype than the individual cell type-specific mutants.

An important consideration in our model is the potential contribution of non-cell autonomous mechanisms to photoreceptor degeneration. *Sox9* is expressed in both MG and RPE cells, and both cell types are known to support photoreceptor viability (*Poché et al., 2008*; *Masuda et al., 2014*). Notably, *Sox9* and *Otx2* cooperate to regulate visual cycle gene expression in the RPE (*Masuda et al., 2014*), and loss of *Otx2* specifically in the adult RPE leads to secondary photoreceptor degeneration through non-cell autonomous mechanisms (*Housset et al., 2013*). However, RPE-specific deletion of *Sox9* does not induce retinal degeneration and in fact results in *Otx2* upregulation (*Masuda et al., 2014*; *Goto et al., 2018*; *Cohen-Tayar et al., 2018*), suggesting that *Sox9* is not an upstream regulator of *Otx2* in this context. Further investigation into the molecular and cellular interactions between MG, RPE, and photoreceptors may help to clarify the indirect pathways contributing to degeneration in the absence of *Sox9*.

To our knowledge, no retinal abnormality has been described in patients with a mutation in *SOX9* (see *Hejtmancik and Daiger, 2020*). This is probably because CD patients are mostly heterozygous for *SOX9* and show early postnatal lethality, so a retinal degeneration phenotype in adulthood was either overlooked or not addressed. However, *SOX9/Sox9* has a complex regulatory landscape (*Bagheri-Fam et al., 2006*; *Despang et al., 2019*), and mutations in its regulatory region have been associated with tissue-specific human phenotypes (*Wunderle et al., 1998*; *Benko et al., 2009*; *Kim et al., 2015*; *Kurth et al., 2009*). In this context, transgenic mice carrying a human enhancer located around 500 kb upstream of *SOX9* revealed enhancer activity in the eye (*Sreenivasan et al., 2017*). Thus, screening of the *SOX9* regulatory region may provide new molecular mechanisms underlying idiopathic cases of retinal degeneration.

Several expression studies have proposed *Sox9* as a marker of epithelial limbal and corneal stem/progenitor cells, with further support from in vitro experiments validating this finding (*Sartaj et al., 2017*; *Parfitt et al., 2015*; *Menzel-Severing et al., 2018*). Through in vivo cell lineage tracing and gene targeting strategies in mice, combined with the analysis of scRNA-seq data, we shed light on the nature, differentiation dynamics, and function of these stem/progenitor *Sox9*-positive cells. Trajectory analysis of limbal and corneal scRNA-seq data revealed that *Sox9* is highly expressed in a cell population characterized by the expression of stem cell markers (*Gpha2*, *Ifitm3*, *Cd63*, and *Krt14*), and that its expression gradually decreases as these cells enter mitosis and differentiate into suprabasal corneal cells. Consistent with this, we observed a substantial proliferation of *Sox9*-expressing descendant cells, leading to a significant increase in their numbers within the first 10 days after the initiation of the TX treatment. Subsequently, these cells originate small clones with a rapid decrease, and finally, they give rise to few, large, radially oriented clones. The fact that *Sox9* was expressed exclusively in epithelial basal cells, while *Sox9*-expressing descendant cells were observed throughout the entire corneal epithelium, indicates that many *Sox9*-expressing descendant cells experience a decline in *Sox9* expression, initiating their differentiation into cells that proliferate, migrate, and differentiate into suprabasal corneal cells, thus contributing to the maintenance of corneal homeostasis.

The persistence of some small clones (including a radial stripe) of *LacZ⁺ Sox9^{△/△}* cells in the limbal and corneal epithelia of *CAGG-CreERTM;Sox9^{flox/flox}; Rosa26-LacZ* mice nevertheless shows that *Sox9* is not absolutely required for the contribution of cells to the limbal and ocular epithelium, but that the lack of *Sox9* significantly impairs the ability of mutant cells to compete with wild-type cells. This competition would explain why, in contrast to the whole-mount *LacZ* analysis, *Sox9*-null cells were not detected in tissue sections of these eyes. A recent preprint (*Rice et al., 2024*) describes a study that confirms the stem-clonogenic potential of *Sox9*-positive limbal epithelial cells. That study also reports that conditional deletion of *Sox9* leads to abnormal corneal epithelial differentiation and squamous

metaplasia in the central cornea. This phenotype was not observed in our study but is explicable because of the mosaic nature of our model, which may allow for rescue of the normal Wnt and Notch signaling by wild-type cells, preventing transdifferentiation (**Menzel-Severing et al., 2018**).

We also observed long-lived circumferential clones that never exited the limbus. These latter clones are similar to those produced by OLB cells, as recently shown in lineage-tracing experiments, which predominantly self-renew and minimally contribute, if at all, to the formation of lineages maintaining the corneal epithelium during homeostasis. However, in response to corneal injury, OLBs proliferate and play an essential role in wound healing (**Altshuler et al., 2021**; **Farrelly et al., 2021**). Consistent with this, we saw that *Sox9* is highly expressed in the OLB cluster. Furthermore, previous studies utilizing an H2B-GFP/K5tTA mouse model have shown that *Sox9* is expressed in a corneal quiescent/slow-cycling cell population (**Parfitt et al., 2015**), and wound healing assay using a human corneal organ culture revealed an increase in the number of SOX9⁺ cells in both activated limbal and re-grown corneal epithelial cells (**Menzel-Severing et al., 2018**). We also observed that the number and extension of the limbal circumferential clones diminished in *Sox9^{Δ/Δ}-LacZ* corneas, indicating that *Sox9* may also control the proliferation of OLB stem cells.

This study highlights how a single transcription factor can regulate gene expression across neighboring cell types, driving distinct genetic programs and biological functions. In the retina, MG cells play a vital role in providing structural and metabolic support. Here, we demonstrate that *Sox9* is essential for the maintenance and survival of MG cells. In this context, previous studies have shown that Notch signaling, which is critical for the maintenance of retinal progenitor cells and the specification of MG cells (**Mills and Goldman, 2017**), regulates *Sox9* expression. Moreover, *Sox9* has been shown to regulate the expression of type I collagen (COL1; **Wang et al., 2013**) and GFAP, a key component of the intermediate filaments that form part of the MG cytoskeleton (**Kinouchi, 2003**). Thus, within the retina, *Sox9* appears to fulfill multiple roles, including the regulation of structural protein expression and the promotion of cell survival through anti-apoptotic activity. This situation closely parallels that shown in postnatal and adult Sertoli cells, which die after *Sox9* ablation, leading to the complete degeneration of the testicular germinative epithelium. In these cells, *Sox9* likewise acts as an anti-apoptotic factor and regulates the expression of extracellular matrix components (**Barrionuevo et al., 2009**; **Barrionuevo et al., 2016**). These facts suggest a role for *Sox9* in the maintenance of multiple adult tissues. In contrast, in the cornea, *Sox9* appears to be involved in the regulation of proliferation and differentiation of corneal epithelial stem/progenitor cells. However, the regulatory mechanisms and downstream targets of *Sox9* in this context remain poorly understood. Here, we show that high *Sox9* expression is associated with the presence of stem cell markers such as *Krt15*, *P63*, and *Gpha2*, whereas its levels decline as stem cells undergo differentiation, consistent with recent findings (**Rice et al., 2024**). Furthermore, in vitro studies using limbal epithelial stem/progenitor cells have demonstrated that activation of the BMP, Notch, and Shh pathways leads to upregulation of *Sox9*, while Wnt signaling antagonizes its activity (**Menzel-Severing et al., 2018**). Thus, in the limbus, unlike its role in the retina, *Sox9* appears to engage with key molecular pathways that govern stem cell maintenance and differentiation. Future studies will help to elucidate the molecular networks in which *Sox9* is involved in both contexts.

In summary, through functional analyses in mice, we have demonstrated multiple roles for *Sox9* in the adult eye, acting as an essential maintenance factor preventing retinal degeneration on the one hand, and promoting the differentiation of limbal cells in the cornea on the other.

## Materials and methods
### Mouse lines and crosses
To obtain *Sox9* null mutant mice, we crossed *Sox9^{flox/flox}* (*B6.129S7-Sox9^{tm2Crm/J}*) to *CAGG-CreER* (*B6.Cg-Tg^{(CAG-cre/Esr1*)5Amc}*) (**Hayashi and McMahon, 2002**) mice, and the resulting double heterozygous offspring were bred to *Sox9^{flox/flox}* mice. Both mouse strains were sourced from The Jackson Laboratory. Mutation occurs after treatment with tamoxifen, as described below. Cre-negative *Sox9^{flox/flox}*, tamoxifen-treated mice served as controls. For corneal *Sox9* mosaic analyses, we crossed *CAGG-CreER^{T2}*; *Sox9^{flox/flox}* mice to *Rosa26-LacZ* (*B6.129S4-Gt(ROSA)26Sor^{tm1Sor/J}*) (**Soriano, 1999**), generating *CAGG-CreER*; *Sox9^{flox/flox}*; *Rosa26-LacZ* mice. As a control, we used *CAGG-CreER*; *Sox9^{flox/+}*; *Rosa26-LacZ* mice. For genetic lineage tracing assays, we used *Sox9^{IRES-CreERT2}* (*B6.129S7-Sox9^{tm1(cre/ERT2)Haak}*)

(*Soeda et al., 2010*) mice obtained from the RIKEN BioResource Research Center (Tsukuba, Japan). These mice were crossed with either *Rosa26-LacZ* (*B6.129S4-Gt(ROSA)26Sor^{tm1Sor/J}*) (*Soriano, 1999*) or *Rosa26-EYFP* (*B6.129X1Gt(ROSA)26Sor^{tm1(EYFP)Cos/J}*) (*Srinivas et al., 2001*) reporter mice, resulting in double heterozygotes. Both of them were provided by The Jackson Laboratory (Bar Harbor, ME). All the experiments started with adult mice (2 months old). Mice were housed under Specific Pathogen-Free conditions in the animal facilities of the Center for Biomedical Research (University of Granada, Granada, Spain). The animals had ad libitum access to food and water and were kept in groups. The occupancy density of the cages (microventilated) was in accordance with legal requirements. The cages were kept at a temperature of 22 ± 2°C, a humidity of 55 ± 10%, and a 12/12 hr dark–light cycle. Mice were provided with activity elements in the form of nest building material and hiding places. The animal procedures and housing conditions were approved by the University of Granada Ethics Committee for Animal Experimentation and the Consejería de Agricultura, Ganadería, Pesca y Desarrollo Sostenible of the Andalusian government, Junta de Andalucía (reference 12/12/2016/177).

## Tamoxifen supplementation

For conditional deletion of *Sox9*, *CAGG-CreERTM;Sox9^{flox/flox}* mice were fed a standard diet (Harlan, 2914) supplemented with tamoxifen (TX; Sigma-Aldrich, St. Louis, MO, C8267) at a concentration of 40 mg TX/100 g diet for 10 days. For short-term tracing of *Sox9*-descendant cells (less than 10 days), TX (Sigma-Aldrich, St. Louis, MO, T5648) was dissolved in corn oil at a concentration of 30 mg/ml and administered at a dose of 0.2 mg per gram of body weight through intraperitoneal injection to *Sox9^{IRES-CreERT2}* mice. For long-term tracing of *Sox9*-positive cells (more than 10 days), *Sox9^{IRES-CreERT2}* mice were administered TX as detailed above for the conditional deletion analyses.

## Estimation of the percentage of tamoxifen-induced, Cre-mediated recombination and GFAP quantification

To estimate the percentage of cells undergoing Cre-mediated recombination in TX-treated *CAGG-CreERTM;Sox9^{flox/flox}* mice, the total number of SOX9+ cells was divided by the total number of SOX8+ cells in a 20x microphotograph of a retinal section stained for SOX9 and SOX8.

In parallel, to quantify GFAP expression as a measure of MG reactivity, we analyzed GFAP immunofluorescence intensity across defined retinal surface areas. Given the cytoplasmic distribution of GFAP within glial processes, direct cell counting was not feasible. Instead, fluorescence intensity was measured using ImageJ, within full-thickness retinal regions in 20x microphotographs of a retinal sections stained for GFAP. The total GFAP signal was normalized to the measured area for each section.

## Histology and immunofluorescence

Eyes were dissected out and the crystalline lenses removed in order to facilitate histological sectioning. The eyes were fixed in 4% paraformaldehyde (PFA), dehydrated in increasing EtOH/saline solutions (EtOH + 0.9% Nacl), embedded in paraffin, sectioned, and either stained with hematoxylin–eosin or processed for protein immunodetection. For simple and double immunofluorescence, sections were incubated overnight with the primary antibodies, washed, incubated with the appropriate conjugated secondary antibodies for 1 hr at room temperature (RT), and counterstained with 40,6-diamidino-2-phenylindole (DAPI). Several sections of eyes from control and mutant mice were consistently mounted on the same slide and processed together. Parallel negative controls were performed in which the primary antibody was omitted. The following antibodies were used: anti-SOX8 (1/500), kindly provided by Dr. Wegner at the Universität Erlangen-Nürnberg (Germany); anti-OPN1SW (1/200, Santa Cruz, sc-14363); anti-OPN1LW (1/500, Chemicon, AB5405); anti-RHO (1/500, Sigma, O4886); anti-BRN3α (1/100, Chemicon, MAB1585); anti-PAX6 (1/40, kindly provided by the Developmental Studies Hybridoma Bank at the University of Iowa), anti-AP2α (1/50, Abcam, ab108311), anti-SOX9 (1/400, Millipore, AB5535), anti-GFP (1/100, Novus Biologicals, NB600-308), anti-p63 (1/500, Master Diagnostica MAD-000479QD), and anti-S100 (prediluted 1/1, Master Diagnostica MAD-000592Q).

The two eyes of each mouse were analyzed in all experiments, and there were no notable differences between them (left–right) either histologically or in the expression pattern of the markers

studied. Similarly, we found no differences between the eyes of control mice regardless of whether they were treated with TX or not.

## Analysis of retinal and corneal cell death

Retinal and corneal cell death was assessed using the TUNEL Fluorescent In Situ Cell Death Detection Kit (Roche, Mannheim, Germany) according to the manufacturer's instructions.

## Retinal flat whole-mount immunofluorescence

Eyes were fixed in 4% PFA and retinas were dissected in TBS containing 0.05% Tween-20 and 0.1% Triton X-100 for 30 min at RT. Retinas were then washed in TBS with 0.05% Tween-20 (TBS-T) and exposed to primary antibodies diluted in PBS with 4% BSA overnight at 4°C. After washing with TBS-T, the retinas were incubated with secondary antibodies diluted in PBS with 4% BSA for 3 hr at RT. Finally, the retinas were mounted flat for confocal microscopy.

## Whole-mount X-gal staining

The eyes were dissected and cut transversely to facilitate solution penetration and whole-mount β-galactosidase histochemical reaction using X-gal as a substrate (X-gal staining) was performed. Whole-mount X-gal staining was performed overnight as described by *Hogan et al., 1995*. After staining, eyes were fixed in 4% paraformaldehyde.

## Estimation of the percentage of LacZ-positive corneal surface area

The surface area was calculated using the FIJI (V2.1.0) program based on ImageJ. The entire cornea surface was outlined, and the areas of the LacZ stripes were manually selected and marked. Then, the percentage of the blue-stained area relative to the total corneal area was calculated. An automatic threshold could not be applied due to variations in brightness across the cornea pictures.

## Imaging

Histological and immunofluorescence images were photographed using a DS-Fi1c camera installed on a Nikon Eclipse Ti microscope (Japan). Retinal flat mount images were obtained using a high-speed spectral confocal microscope Nikon A1 ASHS-1. Subsequently, GIMP2 was used to adjust the color levels and the contrast of the immunofluorescence images. Whole-mount X-gal-stained eyes were photographed using a DP70 digital camera mounted on an Olympus SZX12 stereomicroscope (Japan).

## Single-cell RNA-sequencing dataset analyses

We used the single-cell RNA-sequencing datasets of isolated epithelial cells from the limbus (with marginal conjunctiva and corneal periphery) generated by Altshuler and colleagues (*Altshuler et al., 2021*; Gene Expression Omnibus GSE167992). Bioinformatic analyses were performed with Seurat (version 5.0.3) (*Hao et al., 2021*) following the author guidelines (https://satijalab.org/seurat/). Cells with unusually high and low numbers of unique feature counts or high mitochondrial counts were filtered out. The data were normalized and subsequently scaled. RunPCA was used for principal component analysis, and cells were clustered at a resolution of 0.6. Uniform manifold approximation and projection was used for nonlinear dimensional reduction. Cluster-specific markers were selected with a minimum average log fold change threshold of 0.25 among genes that were expressed in a minimum of 25% of the cells. Cluster identity was assigned using the following markers: Conjunctiva: *Krt17*, *Krt4*, *Krt19*, *Krt6a*, *Krt13*, *Krt8* (and lack of *Krt12*, *Slurp1*, *Ppp1r3c*); cornea: *Krt12*, *Ppp1r3c*, *Slurp*; outer limbus: *Cd63*, *Ifitm3*, *Gpha2*; inner limbus: *Atf3*, *Socs3*, *Mt1*; basal cells: *Itgb1*, *Itgb4*, *Ccnd1*; supra-basal cells: *Cldn4*, *Cdkn1a*, *Dsg1a*; and mitosis: *Mki67*, *Top2a*, *Ccna2*. For differential expression of cells expressing high and low levels of *Sox9*, we selected cells with high *Sox9*-expression levels (≥0.7) and cells with low *Sox9*-expression levels (between 0.7 and 0.15) with the WhichCells and the SetIdent functions of the Seurat package. Differential expression analysis was performed with the FindMarkers function (min.pct = 0.20, logfc.threshold = 0.01). For trajectory analysis, we used the Python Scanpy suite (*Haghverdi et al., 2016*; *Wolf et al., 2018*) following the general PAGA trajectory inference workflow (https://scanpy.readthedocs.io/en/%C2%A0stable/tutorials/trajectories/paga-paul15.html) with a Leiden

resolution of 0.6. Cluster identity was assigned as described above, and the conjunctiva cluster was removed. For pseudotime analysis (*Wolf et al., 2019*), the OLB cluster was chosen as the root. Gene ontology analyses were done with the enrichGO and compareCluster functions of the clusterProfiler R package.

## Statistical analysis

We analyzed datasets derived from distinct experimental conditions using appropriate statistical tests based on preliminary assessments of normality and variance equality. When these conditions were not met, non-parametric approaches were favored for their robustness to data distribution assumptions. Specifically, we employed the Kruskal–Wallis rank-sum test to identify overall differences among group distributions followed by post hoc Dunn's multiple comparison tests with Bonferroni adjustment to pinpoint specific pairwise comparisons that contributed to these differences. When no significant departures from normality or equality of variances were detected, parametric methods were deemed suitable. So, we conducted a standard Student *t*-test to evaluate the hypothesis of equal means between groups. Complete information on these statistical analyses is provided in *Supplementary file 1B–H, K, L*.

## Acknowledgements

This work has been funded by the following institutions: Programa Operativo FEDER Andalucía 2014–2020, Consejería de Economía, Conocimiento, Empresas y Universidad, Junta de Andalucía (Ref. A.BIO.106.UGR18) and MICIU/AEI /10.13039/501100011033 and FEDER, UE, Ref PID2022-139302NB-I00 to RJ and FJB; Ministerio de Ciencia e Innovación, Instituto de Salud Carlos III, European Regional Development Fund through the program 'Una manera de hacer Europa' (Ref. PI23/00335) to MA.

## Additional information

### Funding

| Funder | Grant reference number | Author |
|---|---|---|
| Consejería de Economía, Conocimiento, Empresas y Universidad, Junta de Andalucía | A.BIO.106.UGR18 | Francisco Javier Barrionuevo |
| Ministerio de Ciencia e Innovación | PID2022-139302NB-I00 | Rafael Jiménez |
| Instituto de Salud Carlos III | PI23/00335 | Miguel Alaminos |

The funders had no role in study design, data collection, and interpretation, or the decision to submit the work for publication.

### Author contributions

Alicia Hurtado, Victor López-Soriano, Miguel Lao, M Angeles Celis-Barroso, Pilar Lazúen, Alejandro Chacón-de-Castro, Yolanda Ramírez-Casas, Investigation, Methodology, Writing – review and editing; Miguel Alaminos, Resources, Supervision, Investigation, Methodology, Writing – review and editing; John Martin Collinson, Resources, Formal analysis, Supervision, Investigation, Methodology, Writing – review and editing; Miguel Burgos, Conceptualization, Resources, Data curation, Software, Formal analysis, Supervision, Funding acquisition, Validation, Investigation, Visualization, Methodology, Writing – review and editing; Rafael Jiménez, Francisco Javier Barrionuevo, Conceptualization, Resources, Data curation, Software, Formal analysis, Supervision, Funding acquisition, Validation, Investigation, Visualization, Methodology, Writing – original draft, Project administration, Writing – review and editing; F David Carmona, Conceptualization, Resources, Data curation, Formal analysis, Supervision, Validation, Investigation, Visualization, Methodology, Writing – original draft, Writing – review and editing

## Author ORCIDs
Rafael Jiménez ⓘ https://orcid.org/0000-0003-4103-8219
F David Carmona ⓘ http://orcid.org/0000-0002-1427-7639
Francisco Javier Barrionuevo ⓘ http://orcid.org/0000-0003-2651-1530

## Ethics

The research work presented here complies with all relevant ethical regulations and was first approved by the University of Granada Ethics Committee for Animal Experimentation and then by the regional government, 'Junta de Andalucía' under license number 12/12/2016/177. Research was performed in accordance with the relevant guidelines and regulations dictated by these committees.

Reviewer #1 (Public review): https://doi.org/10.7554/eLife.102337.3.sa1
Reviewer #2 (Public review): https://doi.org/10.7554/eLife.102337.3.sa2
Author response https://doi.org/10.7554/eLife.102337.3.sa3

# Additional files

## Supplementary files

Supplementary file 1. Transcriptomic data and statistical tests performed in this study. (A) Percentage of Cre-mediated *Sox9* inactivation and retinal phenotypes of the analyzed mice. (B) *Figure 1D* statistics. (C) *Figure 1E* statistics. (D) *Figure 1G* statistics. (E) *Figure 2E* statistics. (F) *Figure 2F, G* statistics. (G) *Figure 3B* statistics. (H) *Figure 3D* statistics. (I) Deregulated genes between cells exhibiting high and low expression levels of *Sox9*. (J) Gene ontology analysis using the deregulated genes of I. (K) *Figure 6C* statistics. (L) *Figure 6F* statistics.

MDAR checklist

## Data availability

All data generated or analyzed during this study are included in the manuscript and supporting files.

The following previously published dataset was used:

| Author(s) | Year | Dataset title | Dataset URL | Database and Identifier |
|---|---|---|---|---|
| Altshuler A, Amitai-Lange A, Tarazi N, Dey S, Strinkovsky L, Hadad-Porat S, Bhattacharya S, Nasser W, Imeri J, Ben-David G, Abboud-Jarrous G, Tiosano B, Berkowitz E, Karin N, Savir Y, Shalom-Feuerstein R | 2021 | Discrete limbal epithelial stem cell populations mediate corneal homeostasis and wound healing | https://www.ncbi.nlm.nih.gov/geo/query/acc.cgi?acc=GSE167992 | NCBI Gene Expression Omnibus, GSE167992 |

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
