## [Editor Report · eLife Assessment]

This **useful** study informs the transcriptional mechanisms that promote stem cell differentiation and prevent degeneration in the adult eye. Through inducible mouse mutagenesis, the authors uncover a dual role for a transcription factor (Sox9) in stem cell differentiation and prevention of retinal degeneration. The data at hand **convincingly** support to the main conclusions. The study will be of general interest to the fields of neuronal development and neurodegeneration.

---

## [Referee Report · Reviewer #1 (Public review)]

Summary:

Hurtado et al. show that Sox9 is essential for retinal integrity, and its null mutation causes the loss of the outer nuclear layer (ONL). The authors then show that this absence of the ONL is due to apoptosis of photoreceptors and a reduction in the numbers of other retinal cell types such as ganglion cells, amacrine cells and horizontal cells. They also describe that Müller Glia undergoes reactive gliosis by upregulating the Glial Fibrillary Acidic Protein. The authors then show that Sox9+ progenitors proliferate and differentiate to generate the corneal cells through Sox9 lineage-tracing experiments. They validate Sox9 expression and characterize its dynamics in limbal stem cells using an existing single-cell RNA sequencing dataset. Finally, the authors show that Sox9 deletion causes progenitor cells to lose their clonogenic capacity by comparing the sizes of control and Sox9-null clones. Overall, Hurtado et al. underline the importance of Sox9 function in retinal cells.

Strengths:

The authors have characterized a myriad of striking phenotypes due to Sox9 deletion in the retina and limbal stem cells which will serve as a basis for future studies.

Weaknesses:

Hurtado et al. highlight the importance of Sox9 in the retina and limbal stem cells by describing several affects of Sox9 depletion in the adult eye. However, it is unclear how or where Sox9 precisely acts as a mechanistic investigation of the transcription factor's role in this tissue is lacking.

---

## [Referee Report · Reviewer #2 (Public review)]

Summary:

Sox9 is a transcription factor crucial for development and tissue homeostasis, and its expression continues in various adult eye cell types, including retinal pigmented epithelium cells, Müller glial cells, and limbal and corneal basal epithelia. To investigate its functional roles in the adult eye, this study employed inducible mouse mutagenesis. Adult-specific Sox9 depletion led to severe retinal degeneration, including the loss of Müller glial cells and photoreceptors. Further, lineage tracing revealed that Sox9 is expressed in a basal limbal stem cell population that supports stem cell maintenance and homeostasis. Mosaic analysis confirmed that Sox9 is essential for the differentiation of limbal stem cells. Overall, the study highlights that Sox9 is critical for both retinal integrity and the differentiation of limbal stem cells in the adult mouse eye.

Strengths:

In general, inducible genetic approaches in the adult mouse nervous system are rare and difficult to carry out. Here, the authors employ tamoxifen-inducible mouse mutagenesis to uncover the functional roles of Sox9 in the adult mouse eye.

Careful analysis suggests that two degeneration phenotypes (mild and severe) are detected in the adult mouse eye upon tamoxifen-dependent Sox9 depletion. Phenotype severity nicely correlates with the efficiency of Cre-mediated Sox9 depletion.

Molecular marker analysis provides strong evidence of Mueller cell loss and photoreceptor degeneration.

A clever genetic tracing strategy uncovers a critical role for Sox9 in limbal stem cell differentiation.

Comments on revised submission:

The revised manuscript is very much improved and has addressed all my concerns.

---

## [Author Response]

The following is the authors’ response to the original reviews.

**Reviewer #1 (Public review):**
Hurtado et al. show that Sox9 is essential for retinal integrity, and its null mutation causes the loss of the outer nuclear layer (ONL). The authors then show that this absence of the ONL is due to apoptosis of photoreceptors and a reduction in the numbers of other retinal cell types such as ganglion cells, amacrine cells, and horizontal cells. They also describe that Müller Glia undergoes reactive gliosis by upregulating the Glial Fibrillary Acidic Protein. The authors then show that Sox9+ progenitors proliferate and differentiate to generate the corneal cells through Sox9 lineage-tracing experiments. They validate Sox9 expression and characterize its dynamics in limbal stem cells using an existing single-cell RNA sequencing dataset. Finally, the authors argue that Sox9 deletion causes progenitor cells to lose their clonogenic capacity by comparing the sizes of control and Sox9-null clones. Overall, Hurtado et al. underline the importance of Sox9 function in retinal and corneal cells.Strengths:The authors have characterized a myriad of striking phenotypes due to Sox9 deletion in the retina and limbal stem cells which will serve as a basis for future studies.Weaknesses:Hurtado et al. investigate the importance of Sox9 in the retina and limbal stem cells. However, the overall experimental narrative appears dispersed.(1) The authors begin by characterizing the phenotype of Sox9 deletion in the retina and show that the absence of the ON layer is due to photoreceptor apoptosis and a reduction in other retinal cell types. The authors also note that Müller glia undergoes gliosis in the Sox9 deletion condition. These striking observations are never investigated further, and instead, the authors switch to lineage-tracing experiments in the limbus that seem disconnected from the first three figures of the paper. Another example of this disconnect is the comparison of Sox9 high and Sox9 low populations using an existing scRNA-seq dataset and the subsequent GO term analysis, which does not directly tie in with the lineage-tracing data of the succeeding Sox9∆/∆ experiments.

We thank the reviewer for their thoughtful observations. We would like to clarify the rationale behind the structure of our study and how the different parts are conceptually connected.

Our central aim was to investigate the role of Sox9 in the adult eye. Given that Sox9 has been extensively studied during embryonic development, we specifically chose to use an inducible conditional knockout strategy (CAG-CreERTM) in order to assess its function postnatally, in the adult eye. This approach revealed a severe retinal phenotype, whereas the cornea showed no overt phenotype. A major strength of our experimental design is that it allowed us to examine the role of Sox9 specifically in the adult eye, avoiding confounding effects from embryonic development. Nevertheless, this approach entails an inherent limitation: the mosaic nature of the CAG-CreERTM system leads to substantial variability in both the extent and distribution of Sox9 inactivation among individual animals. We invested considerable effort over extended periods to obtain reliable and biologically meaningful data despite this variability. We did not proceed further because this mosaicism poses a significant limitation when attempting to dissect downstream mechanisms in a consistent and reproducible manner, making it extremely challenging to perform in-depth mechanistic studies.

Regarding the cornea, given the absence of a clear phenotype upon Sox9 deletion, we expanded our investigation by adding lineage-tracing and transcriptomic analyses to better understand Sox9’s potential role in adult limbal epithelial stem cells. These additional experiments provided valuable insight into Sox9 function in the adult cornea, even in the absence of gross morphological changes. Thus, while the retinal and corneal data stem from different experimental approaches, they are unified by a shared goal: understanding the celltype-specific and tissue-specific functions of Sox9 in the adult eye.

To ensure that other readers do not perceive this apparent disconnect, and overstate our conclusions, we have modified the manuscript. In the Introduction section, we have included the main findings from studies conducted to date on the role of Sox9 in the cornea and retina, and we have removed the corresponding section from the Discussion. We believe it is now clear that our study focuses on the role of Sox9 in the adult eye, in contrast to previous studies, which focused on the developing eye.

In the Discussion section, we have added a new paragraph at the beginning and end that explicitly addresses the relationship between the retinal and limbal findings, illustrating how a single transcription factor can play distinct roles in different tissues within the same organ.

Regarding the reviewer’s comment that the scRNA-seq analyses appear disconnected from the lineage-tracing data, we respectfully disagree. These analyses provide independent transcriptional confirmation that Sox9 is a marker of limbal stem cells, reinforcing the conclusions drawn from our in vivo experiments. These approaches are complementary and they converge on the same biological insight: Sox9 marks a population with stem-like properties in the adult limbus. Nevertheless, we acknowledge the reviewer’s concern and have moderated the tone of our statements in the revised version of the manuscript to better reflect the supporting nature of the scRNA-seq data, without overstating its functional implications.

(2) A major concern is that a single Sox9∆/∆ limbal clone has a sufficiently large size, comparable to wild-type clones, as seen in Figure 6D. This singular result is contrary to their conclusion, which states that Sox9-deficient stem cells minimally contribute to the maintenance of the cornea.

We thank the reviewer for this important observation.

Ligand-independent activity of Cre-ER fusion proteins has been repeatedly reported in various mouse models (Vooijs et al., 2001; Kemp et al., 2004; Haldar et al., 2009). This basal recombinase activity is thought to arise from inappropriate nuclear translocation or proteolysis of the Cre-ER fusion protein, leading to low-level recombination even in the absence of tamoxifen. Consistent with this, prior studies using the same CAGG-CreERTM; R26R-LacZ system for clonal analysis in the cornea have observed sparse reporter expression before tamoxifen administration (Dorà et al., 2015).

In line with these findings, we also detected minimal background LacZ staining in Sox9Δ/ΔLacZ corneas (mean surface area: 0.85%; n = 8 eyes). This low-level staining likely reflects recombination events in transient amplifying or more differentiated cells, which are not expected to generate long-lived clones. However, in the rare instance of a large clone, as shown in Figure 6D, we believe that a spontaneous recombination event may have occurred in a bona fide limbal stem cell, giving rise to a sustained contribution. To rigorously address this potential artefact and assess the true contribution of Sox9-deficient stem cells, we conducted a comparative analysis of 8 control (Sox9Δ/+-LacZ) and 5 mutant (Sox9Δ/ΔLacZ) corneas. This analysis revealed a highly significant 8-fold reduction in the LacZpositive surface area in mutant samples (Sox9Δ/+-LacZ: 6.65 ± 1.77%; Sox9Δ/Δ-LacZ: 0.85 ± 0.85%; paired t-test, p = 0.00017; Figs. 6E and F; Table S12).

We chose to include the image of the large clone in the main figure precisely because it does not align with our working hypothesis. We believe that showing such exceptions transparently is scientifically important and may be valuable for other researchers using similar inducible systems. Nonetheless, based on previous literature, the number of samples analyzed, and the statistically significant reduction in clonal contribution, we maintain that the observed phenotype reflects a true biological effect of Sox9 loss, supporting our conclusion that Sox9-deficient stem cells contribute minimally to corneal maintenance. To make that point clearer, we have introduced the following sentence in lines 462-464 of the revised version of the manuscript.

“A possible explanation for this clone may be that spontaneous ligand-independent activity of Cre-ER fusion may have occurred in a bona fide limbal stem cell, as previously reported (Vooijs et al., 2001; Kemp et al., 2004; Haldar et al., 2009, Dorà et al., 2015).”

**Reviewer #2(Public revciew):**
Sox9 is a transcription factor crucial for development and tissue homeostasis, and its expression continues in various adult eye cell types, including retinal pigmented epithelium cells, Müller glial cells, and limbal and corneal basal epithelia. To investigate its functional roles in the adult eye, this study employed inducible mouse mutagenesis. Adult-specific Sox9 depletion led to severe retinal degeneration, including the loss of Müller glial cells and photoreceptors. Further, lineage tracing revealed that Sox9 is expressed in a basal limbal stem cell population that supports stem cell maintenance and homeostasis. Mosaic analysis confirmed that Sox9 is essential for the differentiation of limbal stem cells. Overall, the study highlights that Sox9 is critical for both retinal integrity and the differentiation of limbal stem cells in the adult mouse eye.Strengths:In general, inducible genetic approaches in the adult mouse nervous system are rare and difficult to carry out. Here, the authors employ tamoxifen-inducible mouse mutagenesis to uncover the functional roles of Sox9 in the adult mouse eye.Careful analysis suggests that two degeneration phenotypes (mild and severe) are detected in the adult mouse eye upon tamoxifen-dependent Sox9 depletion. Phenotype severity nicely correlates with the efficiency of Cre-mediated Sox9 depletion.Molecular marker analysis provides strong evidence of Mueller cell loss and photoreceptor degeneration.A clever genetic tracing strategy uncovers a critical role for Sox9 in limbal stem cell differentiation.Weaknesses:(1) The Introduction can be improved by explaining clearly what was previously known about Sox9 in the eye. A lot of this info is mentioned in a single, 3-page long paragraph in the Discussion. However, the current study's significance and novelty would become clearer if the authors articulated in more detail in the Introduction what was already known about Sox9 in retina cell types (in vitro and in vivo).

We appreciate this insightful comment. Following the reviewer`s suggestion, we have reorganized the manuscript to provide a clearer scientific context in the Introduction. Specifically, we have moved the relevant background information on Sox9 in different retinal cell types—previously included in a single, extended paragraph in the Discussion—into the Introduction. This helps to better frame our study within the context of existing knowledge.

Additionally, we have emphasized more explicitly that our work does not focus on embryonic development, as most previous studies on Sox9 have done, but instead investigates its role in the adult mouse retina and limbus/cornea. We believe this represents an important and novel aspect of our study, as the mechanisms of retinal maintenance and limbal stem cell differentiation in the adult have been less extensively studied.

(2) Because a ubiquitous tamoxifen-inducible CreER line is employed, non-cell autonomous mechanisms possibly contribute to the observed retina degeneration. There is precedence for this in the literature. For example, RPE-specific ablation of Otx2 results in photoreceptor degeneration (PMID: 23761884). Have the authors considered the possibility of non-cell autonomous effects upon ubiquitous Sox9 deletion?

Given the similar phenotypes between animals lacking Otx2 and Sox9 in specific cell types of the eye, the authors are encouraged to evaluate Otx2 expression in the tamoxifen-induced Sox9 adult retina.

We appreciate the insightful comment of the reviewer regarding the potential contribution of non-cell autonomous mechanisms to the retinal degeneration observed upon ubiquitous Sox9 deletion. We agree that this is an important consideration, particularly in the context of findings showing that RPE-specific deletion of Otx2 results in secondary photoreceptor degeneration.

However, we would like to emphasize that RPE-specific deletion of Sox9 does not lead to photoreceptor loss or retinal degeneration, as previously shown (Masuda et al., 2014; Goto et al., 2018; Cohen-Tayar et al., 2018) [PMID: 24634209; PMID: 29609731; PMID: 29986868]. In addition, it was shown that Sox9 deletion in the RPE caused downregulation of visual cycle genes but did not compromise photoreceptor integrity or survival. Interestingly, Otx2 expression was found to be upregulated in the absence of Sox9, further supporting the view that Sox9 is not a simple upstream regulator of Otx2 in the adult RPE (Matsuda, 2014). These findings suggest that RPE dysfunction alone cannot account for the severe retinal phenotype we observe in our model.

In our study, we observed that photoreceptor degeneration correlates strongly with the depletion of Sox9 Müller glial cells. Given the well-established supportive and neuroprotective roles of Müller glia, we interpret the retinal degeneration in our model to be primarily a consequence of Müller cell dysfunction (confirmed by the loss of Müller glia markers, such as SOX8 and S100). This interpretation is further supported by previous studies showing that selective ablation of Müller glia can lead to photoreceptor degeneration through cell-autonomous mechanisms (Shen et al., 2012) [PMID: 23136411].

Nevertheless, we agree that this possibility deserves further investigation, and we have acknowledged it in the following paragraph that has been added to the Discussion section (lines 511-523 of the revised ms):

“An important consideration in our model is the potential contribution of non-cell autonomous mechanisms to photoreceptor degeneration. Sox9 is expressed in both MG and RPE cells, and both cell types are known to support photoreceptor viability (Poché et al., 2008; Masuda et al., 2014). Notably, Sox9 and Otx2 cooperate to regulate visual cycle gene expression in the RPE (Masuda et al., 2014), and loss of Otx2 specifically in the adult RPE leads to secondary photoreceptor degeneration through non-cell autonomous mechanisms (Housset et al., 2013). However, RPE-specific deletion of Sox9 does not induce retinal degeneration and in fact results in Otx2 upregulation (Masuda et al., 2014; Goto et al., 2018; Cohen-Tayar et al., 2018), suggesting that Sox9 is not an upstream regulator of Otx2 in this context. Further investigation into the molecular and cellular interactions between MG, RPE, and photoreceptors may help to clarify the indirect pathways contributing to degeneration in the absence of Sox9.”

Consistent with the above, a new citation has been included:

Housset M, Samuel A, Ettaiche M, Bemelmans A, Béby F, Billon N, Lamonerie T. 2013. Loss of Otx2 in the adult retina disrupts retinal pigment epithelium function, causing photoreceptor degeneration. J Neurosci 33:9890–904. doi:10.1523/JNEUROSCI.1099-13.2013.

(3) The most parsimonious explanation for the dual role of Sox9 in retinal cell types and limbal stem cells is that the cell context is different. For example, Sox9 may cooperate with TF1 in photoreceptors, TF2, in Mueller cells, and TF3 in limbal stem cells, and such cell typespecific cooperation may result in different outcomes (retinal integrity, stem cell differentiation). The authors are encouraged to add a paragraph to the discussion and share their thoughts on the dual role of Sox9.

We thank the reviewer for this thoughtful and constructive suggestion. In , we have added a paragraph at the end of the Discussion addressing the potential dual role of Sox9 in the cornea and retina. In this new section, we discuss how Sox9 might exert distinct functions depending on the cellular context, possibly through interactions with different transcriptional partners in specific cell types. This may help explain the contrasting roles of Sox9 in maintaining retinal integrity versus regulating stem cell differentiation in the limbal epithelium.

(4) One more molecular marker for Mueller glial cells would strengthen the conclusion that these cells are lost upon Sox9 deletion.

We thank the reviewer for this constructive suggestion. To reinforce our conclusion that most Müller glial cells are lost following Sox9 deletion, we analysed the expression of S100, a well-established cytoplasmic marker of Müller glia. As S100 is primarily localized to the innermost Müller cell processes and not restricted to cell bodies, direct cell counting was not feasible. Instead, we quantified the S100+ signal intensity across defined retinal surface areas. This analysis revealed a statistically significant reduction in S100 signal in Sox9^Δ/Δ^ retinas compared to controls. These new data, included in the revised Figure 1 (panels F and G), support and extend our previous observations using SOX8, further confirming the loss of Müller glial cells in Sox9-deficient retinas.

We have also modified the manuscript based on this new evidences as follows:

In the Results section, lines 168-177 of the revised ms, we have added the following paragraph: “To independently validate the loss of MG cells in Sox9-deficient retinas, we examined the expression of S100, a cytoplasmic marker that labels the processes of adult Müller cells. In control retinas, strong S100 immunoreactivity was observed across the inner retina, outlining the typical radial projections of Müller glia (Fig. 1F). In contrast, Sox9Δ/Δ retinas with an extreme phenotype exhibited a marked reduction in S100 signal (Fig. 1G). Given the diffuse cytoplasmic localization of S100, we quantified its expression by measuring the fluorescence signal within a defined surface area of the retina. This analysis revealed a statistically significant reduction in S100 signal intensity in mutant samples (including both mild and extreme phenotypes) compared to controls (Fig. 1G; Table S4), further supporting the loss of MG cells upon Sox9 deletion.”

In Methods, line 684 of the revised ms, the anti-S100 antibody reference and its working dilution have been added.

(5) Using opsins as markers, the authors conclude that the photoreceptors are lost upon Sox9 deletion. However, an alternate possibility is that the photoreceptors are still present and that Sox9 is required for the transcription of opsin genes. In that case, Sox9 (like Otx2) may act as a terminal selector in photoreceptor cells. This point is particularly important because vertebrate terminal selectors (e.g., Nurr1, Otx2, Brn3a) initially affect neuron type identity and eventually lead to cell loss.

We perfectly understand the reviewer’s point. However, we believe that the possibility that Sox9 regulates opsin gene expression without affecting photoreceptor survival is very unlikely in our model. The primary evidence comes from the histological analysis shown in Figure 1B, where hematoxylin and eosin staining clearly demonstrates the complete loss of the ONL in Sox9^Δ/Δ^ retinas exhibiting the extreme phenotype. Similarly, DAPI counterstain also evidences the lack of the ONL in many of our immunofluorescence images of these samples. This morphological disappearance of the ONL strongly supports the conclusion that photoreceptor cells are not merely transcriptionally silent but are physically absent.

Furthermore, TUNEL assays in two retinas with a mild phenotype revealed extensive apoptosis within the ONL, suggesting a progressive degeneration process rather than a selective transcriptional effect. While we acknowledge that transcriptional regulation of opsin genes by Sox9 cannot be entirely ruled out, the observed phenotype is more consistent with a structural loss of photoreceptors rather than a change in their molecular identity alone. Therefore, our data support the interpretation that Sox9 is required for photoreceptor survival, likely through non-cell autonomous mechanisms related to Müller glia dysfunction, rather than acting as a terminal selector within photoreceptor cells themselves.

(6) Quantification is needed for the TUNEL and GFAP analysis in Figure 3.

We have quantified the GFAP immunofluorescence signal across defined surface areas of the retina and found a statistically significant increase in GFAP expression in Sox9^Δ/Δ^ mutants compared to controls (Mann-Whitney U test, P = 0.0240; n = 4 controls, 10 mutants). These quantification data are now included in the revised Figure 3.

Regarding the TUNEL assay, although extensive apoptosis was clearly observed in two Sox9<^Δ/Δ^ retinas with a mild phenotype (as shown in Figure 3A), this pattern was not consistent across the full study mouse cohort. Out of 15 mutant samples analyzed (5 of them previously analyzed and 10 additional ones that have been newly analyzed), only two exhibited this pronounced apoptotic pattern. However, in the remaining 13 mutants, we did observe a small but statistically significant increase in the number of TUNEL+ cells compared to controls (zero-inflated Poisson test, P = 0.028, n = 5 controls, 13 mutants). These results are now included in Figure 3 and in Tables S7 and S8.

This pattern likely reflects the transient nature of apoptosis in the degenerative process, which may occur rapidly and thus be difficult to capture consistently at a single time point. Nevertheless, the quantification supports our conclusion that Sox9 loss is associated with increased photoreceptor cell death.

Based on the above, we have included the following paragraphs in the Results section of the manuscript:

In lines 224-252 of the revised ms, the final version of the paragraph is as follows: “Since photoreceptors are absent in severely affected Sox9-mutant retinas, we conducted TUNEL assays to study the role of cell death in the process of retinal degeneration. In control samples (n=5), almost no TUNEL signal was observed in the retina. In contrast, Sox9^Δ/Δ^ mice (n=15) showed numerous TUNEL+ cells, mainly located in the persisting ONL, indicating that photoreceptor cells were dying (Fig. 3A). Although extensive TUNEL staining in the ONL was clearly observed in two Sox9^Δ/Δ^ retinas with mild phenotypes, this pattern was not consistently present across the full cohort. In the remaining 13 mutant retinas, we observed a modest but noticeable increase in the number of apoptotic cells compared to controls (Fig. 3B; Table S7). Despite a high frequency of zero counts (particularly among controls), the difference between groups reached statistical significance when analyzed using a zeroinflated Poisson model (P = 0.028; n = 5 controls, 13 mutants). These findings suggest that photoreceptor apoptosis following Sox9 deletion may occur acutely and within a narrow temporal window, making it challenging to capture the full degenerative process at a single time point”.

Lines 263-269 of the revised ms: “To support these observations quantitatively, we measured GFAP fluorescence intensity across defined retinal surface areas in control and Sox9^Δ/Δ^ mice (Fig. 3D; Table S8). This analysis revealed a statistically significant increase in GFAP signal in mutant retinas compared to controls (Mann-Whitney U test, P = 0.0240; n = 4 controls, 10 mutants). These results are consistent with a progressive gliotic following Sox9 deletion and provide further evidence that MG cells become reactive in the absence of Sox9”.

Similarly, the section “Estimation of the percentage of tamoxifen-induced, Cre-mediated recombination” has been expanded as follows:

Lines 660-665 of the revised ms: “In parallel, to quantify GFAP expression as a measure of MG reactivity, we analyzed GFAP immunofluorescence intensity across defined retinal surface areas. Given the cytoplasmic distribution of GFAP within glial processes, direct cell counting was not feasible. Instead, fluorescence intensity was measured using ImageJ, within full-thickness retinal regions in 20x microphotographs of a retinal sections stained for GAFP. The total GFAP signal was normalized to the measured area for each section”.

(7) Line 269-320: The authors examined available scRNA-Seq data on adult retina. This data provides evidence for Sox9 expression in distinct cell types. However, the dataset does not inform about the functional role of Sox9 because Sox9 mutant cells were not analyzed with RNA-Seq. Hence, all the data that claim that this experiment provides insights into possible Sox9 functional roles must be removed. This includes panels F, G, and H in Figure 5. In general, this section of the paper (Lines 269-320) needs a major revision. Similarly, lines 442-454 in the Discussion should be removed.

We thank the reviewer for this important observation. We agree that the scRNA-Seq dataset used in this section does not include Sox9 mutant cells and therefore does not allow us to assess the consequences of Sox9 loss-of-function. However, we believe that this analysis still provides valuable complementary information. Specifically, it confirms that Sox9 is expressed in a distinct population of limbal stem cells, and that its expression dynamically changes along differentiation trajectories. Although we do not infer causality or phenotypic consequences, the ability to observe how gene expression programs shift as Sox9 is downregulated offers insights into potential transcriptional programs associated with Sox9 activity.

We have carefully revised Lines 269–320 to remove any overinterpretations, and eliminated the corresponding lines in the Discussion (Lines 442–454). However, we have retained Panels G, and H in Figure 5 with updated text that reflect the descriptive nature of these findings, specifically to illustrate that the Sox9-positive cell signature is consistent with a stem cell genetic program, and that when Sox9 is downregulated some gene pathways involved in stem cell differentiation are upregulated.

**Reviewer #1 (Recommendations for the authors):**
Major points(1) Figure 1C shows the proportions of Sox9+ cells that express Sox8 in control, mild and extreme phenotypes. However, as no quantitative classification of mild and extreme phenotypes is reported along with Figure 1A, the large standard deviation for Sox9∆/∆ mild retina might be due to a misclassification of the sample. Therefore, the authors must ascribe each sample to "mild" or "extreme" based on a quantitative metric.

We appreciate the reviewer’s suggestion to clarify the classification criteria used to distinguish “mild” and “extreme” phenotypes in Sox9^Δ/Δ^ retinas. As noted, our classification was based on a qualitative, phenotypic assessment of retinal morphology in hematoxylin/eosin-stained sections. Specifically, retinas were classified as “extreme” when the outer nuclear layer (ONL) was completely absent, and as “mild” when the ONL was present, although often reduced in thickness. This classification reflects the observable structural depletion of the ONL and aligns well with the extent of Sox9 loss in Müller glial cells, as shown in Figure 1. We acknowledge that some variability exists within the “mild” group, likely due to differences in recombination efficiency and the mosaic nature of tamoxifen-induced deletion.

The phenotypic classification of each individual sample is explicitly provided in Supplementary Table S1. We have also added a statement in the Results section clarifying that this classification was based on qualitative histological criteria rather than a numerical threshold.

Lines 104-113 of the revised ms: “We categorized Sox9^Δ/Δ^ retinas into “mild” and “extreme” phenotypes in order to facilitate interpretation of our data. Clasification was based on a qualitative assessment of ONL integrity in histological sections. Specifically, samples were classified as “extreme” when the ONL was completely depleted, and as “mild” when the ONL persisted, albeit variably reduced in thickness. This phenotypic classification reflects observable structural differences rather than a fixed quantitative threshold. Some variability exists within the “mild” group, likely due to differences in recombination efficiency and the mosaic nature of tamoxifen-induced Cre-mediated Sox9 deletion”

(2) The authors infer Sox9 high and Sox9 low groups of limbal stem cells using an existing scRNA-seq dataset. However, an immunohistology-based validation of this difference is missing. Given that limbal stem cells express Sox9, the authors must examine the heterogeneity in Sox9 levels within the Sox8+ population to demonstrate their claim: "...Sox9 expression decreases as transiently amplifying progenitors undergo progressive differentiation from limbal to peripheral corneal cells." in Line 292. Ideally, this must be further validated using differentiation markers corresponding to CB and ILB populations that show lower Sox9 expression according to the pseudotime graph.

To validate the Sox9 expression results obtained with scRNA-seq, we performed double immunofluorescence for Sox9 and P63, the latter expressed by the basal cells of the limbal epithelium, but not by transient amplifying cells covering the corneal surface (Pellegrini et al., 2001, https://www.pnas.org/doi/abs/10.1073/ pnas.061032098). These results can be observed in the new panel 5F. Accordingly we have included a new paragraph in lines 369-396 of the revised version of the ms:

“To validate these results, we decided to closely examine Sox9 expression in the limbus using immunofluorescence. Previous analyses revealed that the outer limbus is approximately 100 μm wide, while the inner limbus is wider, around 240 μm (Altshuler 2021). We observed that in the region corresponding to the OLB, most cells showed strong Sox9 expression. In the area corresponding to the ILB, this immunoreactivity appeared weaker in the basal layer (corresponding to the ILB proper), and no expression was detected in the suprabasal layers (flattened cells; Fig 5F left). Double immunofluorescence for SOX9 and P63, which is expressed in basal cells of the limbal epithelium, but not by transient amplifying cells covering the corneal surface (Pellegrini et al., 2001) revealed that Sox9 expression was restricted to P63-positive cells (Fig 5F right). These observations confirm that Sox9 is expressed in a basal cell population within both the OLB and ILB, and that its expression decreases in differentiated transient amplifying cells. ”

We also have deleted “This expression pattern is consistent with our immunofluorescence observations" from line 356 of the revised ms.

(3) The authors' claim of "...Sox9-null cells cannot survive or proliferate as well as their wildtype neighbors, and are hence outcompeted over time, leading to an essentially wild-type cornea" does not seem very convincing in the light of Fig.6D and S3B where Sox9 deletion can still allow for a large LacZ+ clone. Their claim of wild-type cornea due to out-competing neighbors must be validated by increasing the number of Sox9-null progenitors, which can be tested by administering tamoxifen for a significantly longer duration, leading to a majority Sox9 deficient progenitor population, and then examining limbal and corneal defects.

As previously discussed, we observed only one instance of a large LacZ+ clone across 8 Sox9^Δ/Δ^-LacZ eyes. Based on prior reports of ligand-independent Cre activity (Vooijs et al., 2001; Kemp et al., 2004; Haldar et al., 2009; Dorà et al., 2015), we believe this rare event likely resulted from spontaneous recombination in a bona fide limbal stem cell, independent of tamoxifen administration. For this reason, we do not expect that increasing the dose or duration of tamoxifen would eliminate such rare events. Furthermore, due to the mosaic and highly variable recombination efficiency of the CAGG-CreERTM system in the adult eye (see McMahon et al., 2008), attempting to increase the TX dosage would likely lead to systemic toxicity or lethality, without guaranteeing full inactivation of the gene in the limbus. Thus, this system is not well-suited for generating a fully Sox9-deficient limbal epithelium. To overcome this limitation, we crossed our mice with the R26R-LacZ reporter line to track the clonal behavior of Sox9-deficient cells. In control animals (Sox9Δ/+-LacZ), LacZ+ stripes originating from limbal stem cells are readily observed. In contrast, in Sox9Δ/Δ-LacZ mutants, these clones are either absent or drastically reduced. This suggests that Sox9-null cells have a severely impaired ability to form and sustain clones. To rigorously quantify this effect, we compared 8 control and 5 mutant corneas, revealing a highly significant 8-fold reduction in LacZ-positive area in the mutants (6.65 ± 1.77% vs. 0.85 ± 0.85%; p = 0.00017; Fig. 6F; Table S12; Supp. Fig. X???), supporting our claim that Sox9null cells cannot survive or proliferate as well as their wild-type neighbors, and are hence outcompeted over time, leading to an essentially wild-type cornea.

Minor points(1) Quantification for Figure 2C and 2D is missing.

We have now included quantification of BRN3A+ retinal ganglion cells (Figure 2E) across control and Sox9^Δ/Δ^ retinas. Cell counts were performed on matched retinal sections, and the difference between groups was found to be statistically significant through Mann–Whitney U test (Table S5).

Regarding PAX6/AP2a, we quantified inner retinal neurons by analyzing AP2α+ amacrine cells and PAX6+/AP2α- horizontal cells as distinct subpopulations, rather than simply comparing total PAX6 or AP2α immunoreactivity. This approach allowed us to better resolve specific neuronal subtype changes. Both populations showed a statistically significant reduction in Sox9-deficient retinas relative to controls. The quantification for these analyses has now been incorporated into the revised Figure 2F and G (Table S6).

Consequently with the above, the following paragraph of the Results section line 210 of the revised ms:

“We also studied the status of other retinal cell types. The transcription factor BRN3A was used to identify ganglion cells (Nadal-Nicolás et al., 2009), which were shown to decrease in number in the mutant retinas, compared to control ones (Fig. 2C). Similarly, double immunodetection of the transcription factors PAX6 and AP2A was used to identify both amacrine and horizontal cells, as previously described (Marquardt et al., 2001; Barnstable et al., 1985; Edqvist and Hallböök, 2004), showing a similar reduction in both cell types in degenerated retinas (Fig. 2D).”

Has been modified as follows:

“We also studied the status of other retinal cell types. The transcription factor BRN3A was used to identify ganglion cells (Nadal-Nicolás et al., 2009), which were shown to decrease in number in the mutant retinas, compared to control ones (Figs. 2C and 2D and Table S5; n = 5 controls, n = 12 mutants; Mann-Whitney U test, P = 3 × 10^-4^). Similarly, double immunodetection of the transcription factors PAX6 and AP2A was used to identify both amacrine and horizontal cells (Fig. 2E), as previously described (Marquardt et al., 2001; Barnstable et al., 1985; Edqvist and Hallböök, 2004), showing a similar reduction in both cell types in degenerated retinas (Figs. 2F and 2G and Table S6; AP2α+ amacrine cells: n = 3 controls, n = 8 mutants; 2-sample T-tests P = 0.029; PAX6+/AP2α− horizontal cells: n = 3 controls, n = 8 mutants; Mann-Whitney U test P = 0.021). These findings indicate that the loss of Sox9 in the adult retina ultimately leads to the degeneration of multiple inner retinal neuronal populations, beyond the previously described effects on photoreceptors and Müller glia.

(2) Figure 4G & H: The authors must mention that the dashed lines enclose the limbal area.

Done

(3) The authors infer from an existing scRNA-seq dataset that OLB cells have high Sox9 expression as compared to ILB and corneal populations. However, Figures 4A and B do not indicate the anatomical positions of these cell types. The authors must label these for the reader's reference as they state that "[Sox9] expression pattern is consistent with our immunofluorescence observations" in Line 280.

As previously indicated, we have generated a new panel 5F and a corresponding paragraph to illustrate Sox9 expression pattern in the limbus. Accordingly, we have removed the sentence from line 280.

(4) Quantification for Figures 6A and 6B is missing.

We have quantified the number of Sox9 and P63 positive cells in the limbus between mutant and control corneas and found no difference in the number of positive cells. We have included these data in new panel 6C and Table S11.

**Reviewer #2 (Recommendations for the authors):**
Line 24: "synapsis" should be "synapses".

Done

(1) Consider starting a new paragraph after line 30.

Done

(2) Lines 42-48: make clear that this paragraph provides information only for HUMAN SOX9.

We now distinguish which studies were conducted in humans and which in mice.

(3) Line 55: explain to the non-expert reader what the "visual cycle" is.

Done (lines 64-65 of the revised ms)

(4) Line 66: consider "inactivate" instead of "suppress".

We substituted “suppress” with “inactivate”

(5) Line 90-92: ONLY PCR for the cGMP will provide formal evidence that this is not present in the mouse line.

We agree with the reviewer that PCR genotyping is the most straightforward method to exclude the presence of the Pde6^brd^1 allele. Although retinal degeneration was never observed in untreated or control animals in our study, we have now removed the term “formal possibility” from the text to better reflect this limitation.

We have modified the following paragraph (page 116 in the revised version of the manuscript): “Retinal degeneration was never observed in mice that had not been tamoxifen-treated, nor any other controls, eliminating the formal possibility that the retinal degeneration allele of photoreceptor cGMP phosphodiesterase 6b (Pde6brd1) was present in our mice (Bowes et al., 1990).”

As follows: “Retinal degeneration was never observed in mice that had not been tamoxifentreated, nor any other control groups, making the presence of the retinal degeneration allele of photoreceptor cGMP phosphodiesterase 6b (Pde6^brd1^) unlikely in our mice (Bowes et al., 1990). However, we acknowledge that definitive exclusion of this possibility would require PCR-based genotyping.”

(6) Line 160-166: This paragraph needs a conclusion.

We agree with the reviewer and have added the following sentence at the end of the paragraph:

“These findings indicate that the loss of Sox9 in the adult retina ultimately leads to the degeneration of multiple inner retinal neuronal populations, beyond the previously described effects on photoreceptors and Müller glia”

(7) Line: 240-265: This paragraph ends without a conclusion.

We have include the following conclusion:

“Thus, Sox9 is expressed in a basal limbal stem cell population with the ability to form two types of long-lived cell clones involved in stem cell maintenance and homeostasis.”

(8) In Results, it needs to be specified when exactly in adulthood the tamoxifen treatment started. This information is only provided in the Methods.

We have specified the age of the mice at the onset of tamoxifen treatment (two months) and included it in the schemes of Figs 1A, 4C, 4H, 6E.

(9) Line 250: Because live imaging is not conducted, the word "dynamics" is not suitable.

We substituted “dynamics” with “contribution”

(10) Panel C in Figure 6 is nice and helpful. Consider adding a similar panel in Figure 1.

Done.

(11) Line 420: is this the human Sox9 enhancer?

Yes. It is a human enhancer. We have indicated it in the text.

(12) Line 459: typo "detected tissue".

Corrected

(13) Line 448 and 468: citations are needed.

Line 448 is deleted in the revised version of the ms.

(14) 479: typo "clones clones'.

Corrected.